# Hemi-methylated DNA regulates DNA methylation inheritance through allosteric activation of H3 ubiquitylation by UHRF1

Joseph S Harrison[1,2], Evan M Cornett[3†], Dennis Goldfarb[4†], Paul A DaRosa[5†], Zimeng M Li[6], Feng Yan[7], Bradley M Dickson[3], Angela H Guo[1], Daniel V Cantu[1], Lilia Kaustov[8], Peter J Brown[8], Cheryl H Arrowsmith[8], Dorothy A Erie[9], Michael B Major[4,7], Rachel E Klevit[5], Krzysztof Krajewski[1], Brian Kuhlman[1,2], Brian D Strahl[1,2], Scott B Rothbart[3*]

[1]Department of Biochemistry and Biophysics, University of North Carolina at Chapel Hill, Chapel Hill, United States; [2]Lineberger Comprehensive Cancer Center, University of North Carolina at Chapel Hill, Chapel Hill, United States; [3]Center for Epigenetics, Van Andel Research Institute, Grand Rapids, United States; [4]Department of Computer Science, University of North Carolina at Chapel Hill, Chapel Hill, United States; [5]Department of Biochemistry, University of Washington, Seattle, United States; [6]Department of Physics and Astronomy, University of North Carolina at Chapel Hill, Chapel Hill, United States; [7]Department of Cell Biology and Physiology, University of North Carolina at Chapel Hill, Chapel Hill, United States; [8]Structural Genomics Consortium, University of Toronto, Toronto, Canada; [9]Department of Chemistry, University of North Carolina at Chapel Hill, Chapel Hill, United States

*For correspondence: scott.
rothbart@vai.org

†These authors contributed
equally to this work

Competing interests: The
authors declare that no
competing interests exist.

Reviewing editor: Jerry L
Workman, Stowers Institute for
Medical Research, United States

**Abstract** The epigenetic inheritance of DNA methylation requires UHRF1, a histone- and DNA-binding RING E3 ubiquitin ligase that recruits DNMT1 to sites of newly replicated DNA through ubiquitylation of histone H3. UHRF1 binds DNA with selectivity towards hemi-methylated CpGs (HeDNA); however, the contribution of HeDNA sensing to UHRF1 function remains elusive. Here, we reveal that the interaction of UHRF1 with HeDNA is required for DNA methylation but is dispensable for chromatin interaction, which is governed by reciprocal positive cooperativity between the UHRF1 histone- and DNA-binding domains. HeDNA recognition activates UHRF1 ubiquitylation towards multiple lysines on the H3 tail adjacent to the UHRF1 histone-binding site. Collectively, our studies are the first demonstrations of a DNA-protein interaction and an epigenetic modification directly regulating E3 ubiquitin ligase activity. They also define an orchestrated epigenetic control mechanism involving modifications both to histones and DNA that facilitate UHRF1 chromatin targeting, H3 ubiquitylation, and DNA methylation inheritance.

## Introduction

Epigenetic regulation of chromatin architecture and gene expression is driven, in large part, by proteins that write, erase, and read histone post-translational modifications (PTMs) and DNA methylation. These proteins and their complexes are often comprised of multiple regulatory domains, permitting intricate mechanisms that govern allosteric control of enzymatic activity and multivalent engagement of chromatin through one or more reader modules (*Du et al., 2015*; *Musselman et al., 2012*; *Noh et al., 2016*; *Rothbart and Strahl, 2014*; *Ruthenburg et al., 2007*; *Su and Denu, 2016*).

**eLife digest** Cells are able to regulate the activity of their genes in response to different cues. Genetic information is encoded in DNA and one way to regulate gene activity is to modify the DNA by attaching chemical "epigenetic" markers to it. When a cell divides, these epigenetic markers can be inherited by the daughter cells so that they share the same patterns of gene activity as the parent cell. When the DNA of the parent cell is copied prior to cell division, the epigenetic markers are also copied onto the new DNA. Mistakes in this process are linked to a wide range of diseases in humans, such as cancer and neurological disorders.

One type of epigenetic marker is known as a methyl tag and it is added to DNA by certain enzymes in a process called DNA methylation. A protein called UHRF1 is required for human cells to inherit patterns of DNA methylation through cell division. This protein binds to newly copied DNA that lacks some methyl tags as well as to another protein associated with DNA called histone H3. UHRF1 modifies histone H3 by attaching a small protein molecule called ubiquitin to it. This helps to recruit a DNA methylation enzyme to place methyl tags on the newly copied DNA. However, it was not clear how the various properties of UHRF1 allow it to control how DNA methylation is inherited.

Harrison et al. addressed this question by studying purified proteins and DNA fragments outside of living cells. The results show that UHRF1 binding to DNA and histone H3 work together to bring UHRF1 to the sites on DNA that require methylation. Further experiments revealed that the methylation pattern on newly copied DNA is able to activate the ability of UHRF1 to place ubiquitin on histone H3.

The findings of Harrison et al. reveal a new mechanism by which dividing cells control how DNA methylation is inherited by their daughter cells. A future challenge will be to find out how attaching ubiquitin to histone H3 activates DNA methylation.

The E3 ubiquitin ligase UHRF1 (ubiquitin-like, containing PHD and RING finger domains 1) is one such multi-domain epigenetic regulator (*Figure 1A*) that plays a central role in DNMT1-directed DNA methylation maintenance during DNA replication (*Bostick et al., 2007*; *Sharif et al., 2007*). It does so in part through the reader activity of its linked TTD-PHD (tandem Tudor and plant homeodomain) towards the N-terminus of histone H3 when it is di- and tri-methylated at lysine 9 (H3K9me2/me3) (*Arita et al., 2012*; *Rothbart et al., 2013*, *2012*), and through RING (really interesting new gene) domain-mediated catalysis of H3K18 and H3K23 ubiquitylation that promotes DNMT1 association with H3 (*Nishiyama et al., 2013*; *Qin et al., 2015*).

The SRA (SET and RING-associated domain) of UHRF1 binds DNA with modest selectivity towards hemi-methylated CpG dinucleotides (HeDNA) (*Arita et al., 2008*; *Avvakumov et al., 2008*; *Hashimoto et al., 2008*) and has also been implicated in DNA methylation regulation. However, as previously studied mutations and deletions of the SRA disrupt DNA interaction regardless of DNA methylation status (*Liu et al., 2013*; *Sharif et al., 2007*), the specific contribution of HeDNA recognition to this epigenetic regulatory process has not been defined. We therefore sought to gain insight into the function of HeDNA recognition through the UHRF1 SRA domain and determine the relationship between the enzymatic and histone- and DNA-binding activities of this multi-domain epigenetic regulator.

## Results

We first produced recombinant full-length human UHRF1 and quantified the interaction of this protein with double-stranded DNA oligonucleotides containing a single unmodified (UnDNA), hemi-methylated (HeDNA), or symmetrically methylated (SyDNA) CpG dinucleotide by fluorescence polarization (FP). UHRF1 displayed a 10- to 20-fold preference for HeDNA over UnDNA, and a 5- to 10-fold preference for HeDNA over SyDNA (*Figure 1B*; left panel). We also confirmed the binding preferences of two previously characterized single amino acid substitutions to the SRA domain (*Avvakumov et al., 2008*). G448D (DNA^mut^) disrupts all DNA-binding (*Figure 1B*; middle panel) by installing a negatively charged residue at a position that contacts the DNA backbone (*Figure 1—figure supplement 1*), and N489A (HeDNA^mut^), harbored within the NKR finger that contacts the

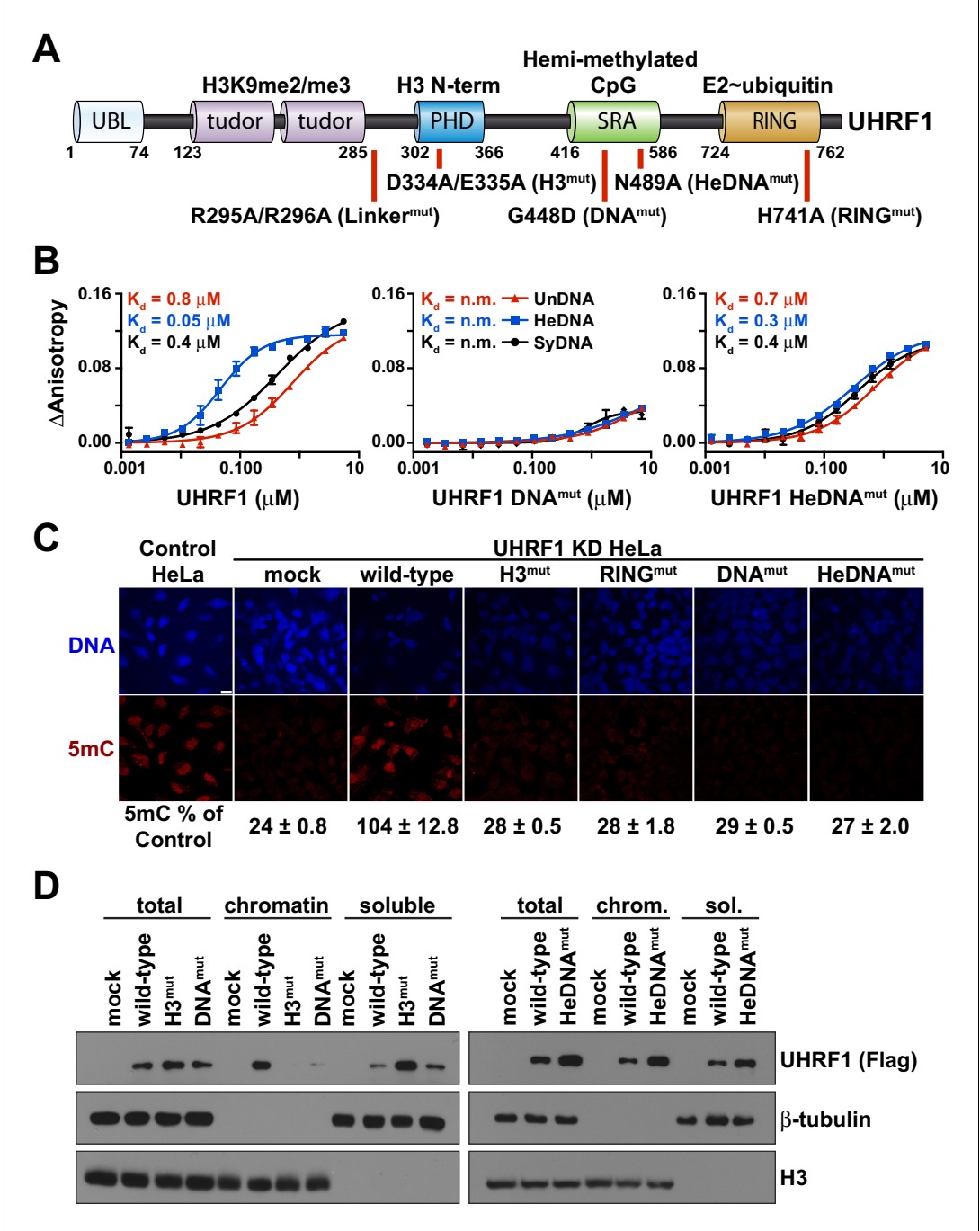

**Figure 1.** UHRF1 binding to HeDNA is required for DNA methylation regulation but is dispensable for chromatin interaction. (**A**) Domain map of human UHRF1 with identified biochemical functions (top) and loss-of-function point mutations used in this study (bottom; see also *Figure 1—figure supplement 1*). UBL (ubiquitin-like); TTD (tandem Tudor domain); PHD (plant homeodomain); SRA (SET and RING-associated domain); RING (really interesting new gene). Amino acid positions demarcating domain boundaries are also shown. (**B**) FP binding assays quantifying the interaction of wild-type, DNA^mut, and HeDNA^mut MBP-tagged UHRF1 with the indicated FAM-labeled DNA oligonucleotides. Error is represented as ± s.e.m. for two independent experiments. (**C**) Representative immunofluorescence staining for 5-methylcytosine (5mC) in control and UHRF1 knockdown Hela cells after genetic complementation with the indicated wild-type and mutant forms of full-length UHRF1. Error is represented as ± S.D. from at least four fields of view. Mock, no DNA control; Scale bar, 20 µm. (**D**) Chromatin association assays for FLAG-tagged UHRF1 (wild-type) or the indicated mutants from asynchronously growing HeLa cells. Mock, no DNA control.

The following figure supplements are available for figure 1:

**Figure supplement 1.** UHRF1 mutations characterized in this study.

*Figure 1 continued*

**Figure supplement 2.** The DNA binding affinity of UHRF1 is highly sensitive to salt concentration.

unmethylated cytosine opposite the methylated base (*Figure 1—figure supplement 1*), disrupts only HeDNA-sensing (*Figure 1B*; right panel). We also observed that the DNA binding affinity of UHRF1 was exquisitely sensitive to small perturbations in salt concentration; we measured a nearly 500-fold affinity difference for HeDNA between 50 mM and 150 mM NaCl (*Figure 1—figure supplement 2*).

We next used a previously developed genetic complementation system in HeLa cells (*Rothbart et al., 2012*) to determine the contribution of DNA-binding and HeDNA-sensing to UHRF1 function in DNA methylation maintenance. Consistent with our previous observations (*Rothbart et al., 2013*, *2012*), global DNA methylation levels were significantly reduced following stable knockdown of endogenous UHRF1 by shRNA (*Figure 1C*). DNA methylation was restored by reintroduction of a wild-type UHRF1 transgene, but like mutations that disrupt histone interaction through the PHD finger (H3$^{mut}$; *Figure 1—figure supplement 1*), E3 ubiquitin ligase activity (RING-$^{mut}$) and DNA binding (DNA$^{mut}$), HeDNA$^{mut}$ could not rescue DNA methylation loss in cells despite retaining its ability to bind DNA (*Figure 1B–C*). These results demonstrate that in addition to the well-appreciated roles of histone-binding and ubiquitin ligase activity to the DNA methylation regulatory function of UHRF1 (*Nishiyama et al., 2013*; *Rothbart et al., 2013*, *2012*), hemi-methylated DNA sensing is critical for DNA methylation maintenance.

Notably, unlike H3$^{mut}$ and DNA$^{mut}$, wild-type and HeDNA$^{mut}$ bound to bulk chromatin biochemically fractionated from HeLa cells (*Figure 1D*). Collectively, these findings suggest that the histone- and DNA-binding domains of UHRF1 are performing complementary functions to target UHRF1 to chromatin, and that HeDNA recognition provides an additional regulatory layer in the DNA methylation program.

To test this hypothesis, we first sought to determine whether the independently characterized DNA- and histone-binding activities of UHRF1 might function in concert. In agreement with previous analyses of the isolated TTD-PHD (*Rothbart et al., 2013*, *2012*), full-length UHRF1 displayed a preference for H3K9me3 peptides over unmodified H3 peptides (H3K9un) (*Figure 2A*; top panel, see also *Supplementary file 1*). No binding was observed for H3$^{mut}$ (*Figure 2—figure supplement 1A*) or for wild-type protein binding to peptides containing an N-terminal 5-carboxyfluorecin (FAM) probe to block PHD engagement (*Figure 2A*; bottom panel). Performing these assays with full-length UHRF1 allowed us to ask whether DNA binding affects H3 peptide binding and vice versa. Histone binding measurements in the presence of 10 µM unlabeled HeDNA, SyDNA, or UnDNA enhanced the interaction with C-terminal FAM-labeled H3K9me3 and H3K9un peptides (*Figure 2A*; top panel). HeDNA did not enhance binding to N-terminal FAM-labeled peptides, indicating that the multivalent interaction of the TTD-PHD with a single H3 peptide (*Rothbart et al., 2013*, *2012*) remained intact. Reciprocally, DNA-binding measurements in the presence of 10 µM unlabeled H3K9me3 peptide enhanced DNA binding affinity 5–10 fold irrespective of the methylation status on DNA (*Figure 2B–C* and *Figure 2—figure supplement 1B-C,E*). Collectively, these experiments demonstrate that the histone- and DNA-binding modules of UHRF1 are regulated by reciprocal positive allostery.

In agreement with the multivalent histone engagement model of the UHRF1 TTD-PHD, the extent to which H3 peptides augmented the interaction of UHRF1 with DNA was dependent on the epigenetic signature on H3. H3K9me3 peptide showed a three-fold enhancement of DNA binding over H3K9un, and asymmetric di-methylation of arginine 2 (H3R2me2a), which blocks the UHRF1 PHD interaction with H3 (*Rajakumara et al., 2011*), did not enhance DNA binding (*Figure 2B*). Consistently, H3$^{mut}$ completely perturbed the ability of an H3K9me3 peptide to positively regulate DNA binding, a previously characterized double mutation to the linker connecting the TTD-PHD that uncouples multivalent engagement to H3 (Linker$^{mut}$) (*Arita et al., 2012*; *Rothbart et al., 2013*) exhibited a weaker enhancement of DNA binding in the presence of peptide than wild-type, and DNA$^{mut}$ remained unable to bind DNA in the presence of H3K9me3 (*Figure 2C* and *Figure 2—*

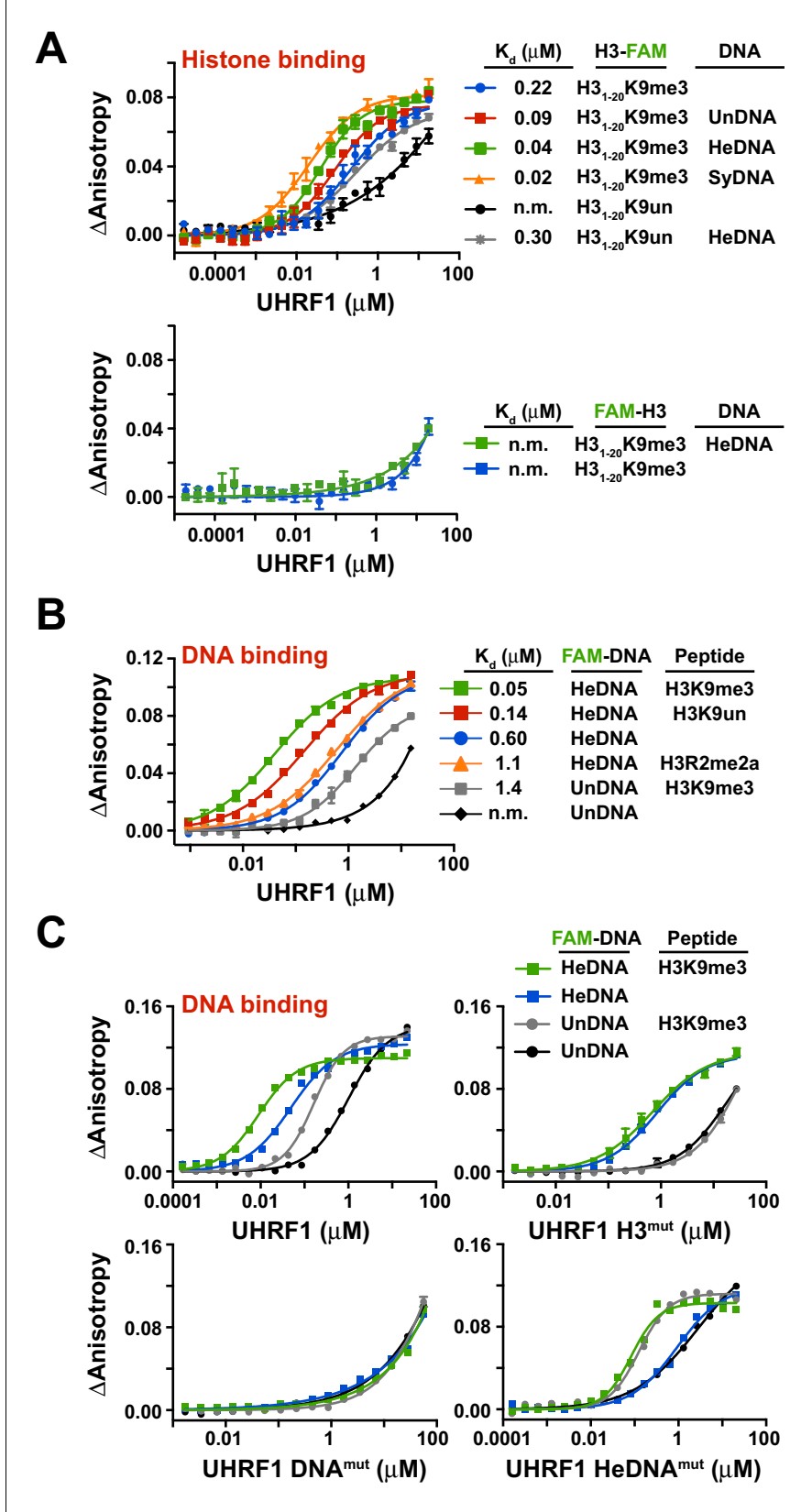

**Figure 2.** The DNA- and histone-binding domains of UHRF1 are regulated by reciprocal positive allostery. (**A**) FP binding assays quantifying the interaction of MBP-UHRF1 with a C-terminally FAM-labeled H3$_{1-20}$K9me3 peptide

*Figure 2 continued on next page*

*Figure 2 continued*

(see *Supplementary file 1* for a full list of peptides used in this study) in the absence or presence of the indicated unlabeled DNA oligonucleotides. Error is represented as ± s.e.m. for two independent experiments. (**B–C**) FP binding assays quantifying the interactions of wild-type and the indicated mutant MBP-UHRF1 proteins with FAM-labeled HeDNA or UnDNA in the presence and absence of the indicated unlabeled H3$_{1-20}$ peptides. Error is represented as ± s.e.m. for two independent experiments. See *Figure 2—figure supplement 1E* for K$_d$ values associated with panel C.

The following figure supplement is available for figure 2:

**Figure supplement 1.** Quantifying the interaction of full-length UHRF1 and various mutants with histone H3 peptides and DNA oligonucleotides.

---

*figure supplement 1D–E*). Conversely, the DNA binding affinity of HeDNA$^{mut}$ was still enhanced by H3K9me3, although HeDNA$^{mut}$ could not discriminate between UnDNA and HeDNA (*Figure 2C*), providing a biochemical basis for HeDNA$^{mut}$ retention on chromatin (*Figure 1D*).

The observed positive allostery between DNA- and histone-binding suggested the possibility of a direct physical interaction between the SRA and TTD-PHD domains of UHRF1. Consistent with this hypothesis, the UHRF1 TTD-PHD associated with the SRA and SRA-RING in pull-down experiments (*Figure 3A*), and this association was perturbed in the presence of DNA, irrespective of methylation status (*Figure 3B*, left). SRA-RING DNA$^{mut}$ maintained interaction with the TTD-PHD in the presence of DNA (*Figure 3B*, right). However, an H3K9me2 peptide did not inhibit the interaction between the SRA and the TTD-PHD (*Figure 3C*). These results suggest that the DNA-binding surface of the SRA contributes to an intramolecular interaction in a manner non-competitive with histone binding. To ensure that the allostery observed was due to an intramolecular rearrangement and not through oligomerization, we characterized UHRF1 in the presence and absence of ligands with several bio-physical techniques. Indeed, UHRF1 remained monomeric and in good agreement with the expected molecular weight as measured by analytical size exclusion chromatography, dynamic light scattering, and atomic force microscopy (*Figure 3D–F*).

Collectively, these results show that the histone- and DNA-binding domains of UHRF1 interact and that general DNA binding releases this physical association. The data further suggest that ligand-induced intramolecular rearrangement of UHRF1 domain connectivity results in high-affinity retention of UHRF1 on chromatin through positive regulation of histone- and DNA-binding activities. This model is generally consistent with a recent report published during the preparation of this man-uscript (*Fang et al., 2016*), which shows that HeDNA enhances histone interaction and suggests a closed-to-open conformational change in UHRF1 intramolecular architecture upon ligand binding. However, two key differences between our findings are that we show reciprocal positive allostery between the histone- and DNA-binding domains of UHRF1, and that general DNA interaction (regardless of methylation state) can displace the TTD-PHD domain and enhance histone interaction. We note that the conclusions from Fang *et al.* relied upon the interpretation of qualitative in-solution pull-down experiments conducted with a 2:1 DNA:UHRF1 ratio (see *Fang et al., 2016*), whereas we used quantitative FP to measure relative binding affinities and included unlabeled ligands in our experiments at concentrations at least five-fold over their measured K$_d$ values to ensure saturation.

Since UHRF1 ubiquitin ligase activity is required to support DNA methylation yet is dispensable for bulk chromatin interaction (see *Figure 1C–D* and *Nishiyama et al., 2013*; *Qin et al., 2015*), we hypothesized that there may be a functional link between HeDNA-binding and UHRF1 ligase activity. To begin testing this hypothesis, we *in vitro* reconstituted UHRF1-mediated ubiquitylation using recombinant UHRF1, H3 peptides, Flag-tagged ubiquitin, and the ubiquitin conjugation enzymes E1 (Uba1) and E2 (UbcH5c) (for a review of the mechanism of ubiquitin activation see *Schulman, 2011*). Surprisingly, we observed robust ubiquitylation of an H3$_{1-32}$K9me2 peptide (*Supplementary file 1*) in a 20 min end-point assay in the presence of HeDNA (*Figure 4A*). Neither apo-UHRF1, SyDNA, nor UnDNA could stimulate this activity at concentrations well above their measured K$_d$ values (*Figure 4A*), despite the ability of these DNAs to positively regulate histone binding (*Figure 2A*). Consistent with our measured K$_d$ for HeDNA (*Figure 1B*), we observed reduced ubiquitylation activity as HeDNA concentration fell below 300 nM (*Figure 4A*). To our knowledge, this is the first

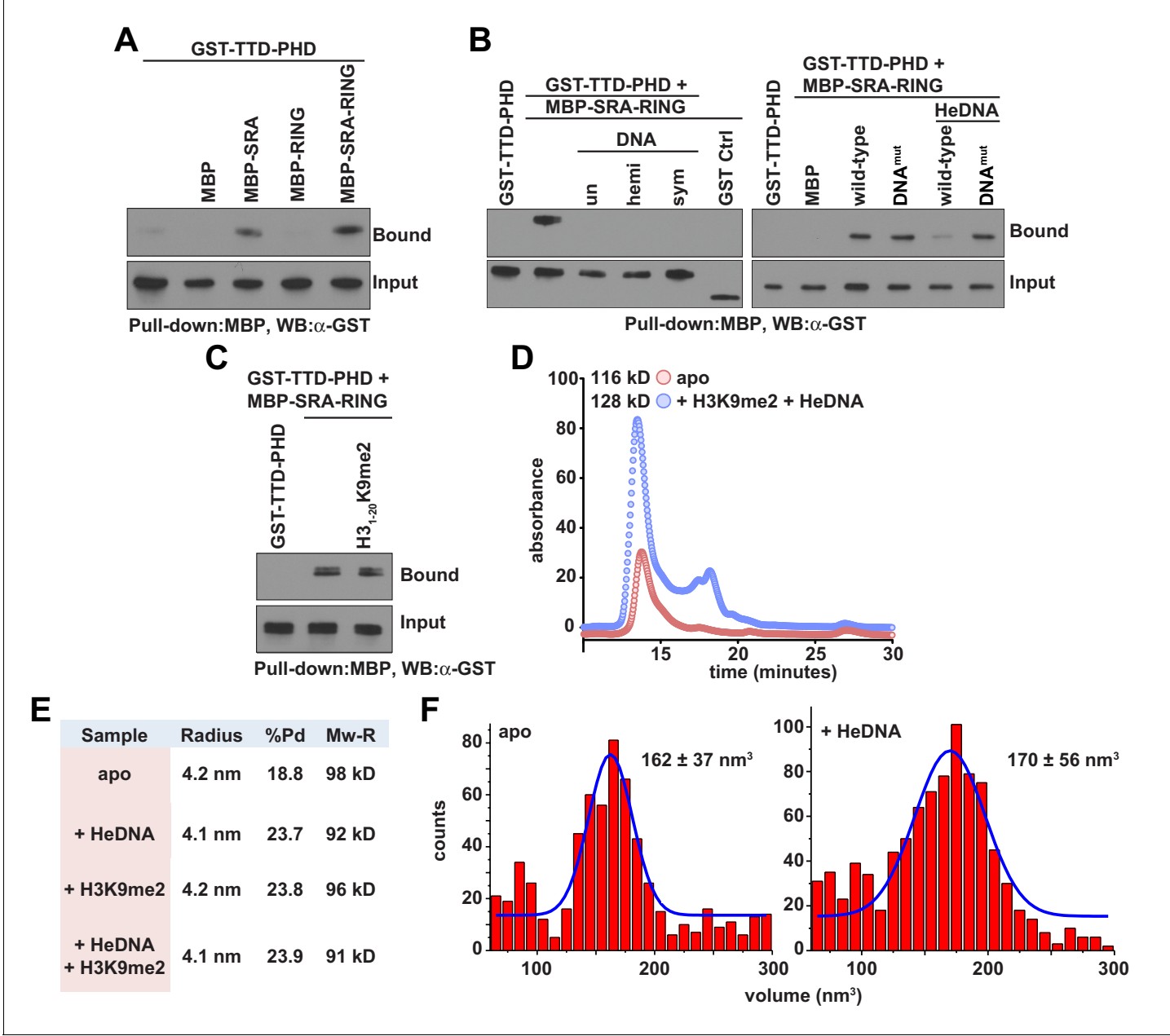

**Figure 3.** DNA binding disrupts a UHRF1 intramolecular interaction. (**A**) In vitro pull-down analysis of the interaction between GST-TTD-PHD and MBP or the indicated MBP fusions of UHRF1. (**B**) Pull-down analysis of the interaction between GST-TTD-PHD and MBP-SRA-RING (wild-type or DNA$^{mut}$) fusions of UHRF1 in the presence or absence of the indicated DNA oligonucleotides. GST Ctrl is a GST fusion of the PHD-Bromo from BPTF (see Materials and methods). (**C**) Pull-down in the presence of H3$_{1-20}$K9me2. (**D**) Analytical size exclusion chromatography of UHRF1 in the absence or presence of HeDNA and H3$_{1-15}$K9me2. The calculated molecular weights for apo and ligand-bound UHRF1 are in agreement with the expected molecular weight of monomeric UHRF1. (**E**) Dynamic light scattering of UHRF1 in the absence or presence of HeDNA and H3$_{1-15}$K9me2. UHRF1 remains mono-dispersed (poly-dispersity < 25%) both in the presence and absence of the indicated ligands. The calculated mass range of 91–98 kD is in agreement with the expected molecular weight for full-length monomeric UHRF1, 90 kD. (**F**) Atomic force microscopy histograms of the volumes for 617 apo UHRF1 particles (left) and 884 HeDNA-bound UHRF1 particles (right). Distributions were fit to a single Gaussian peak using the peak fit function in Origin 6.1.

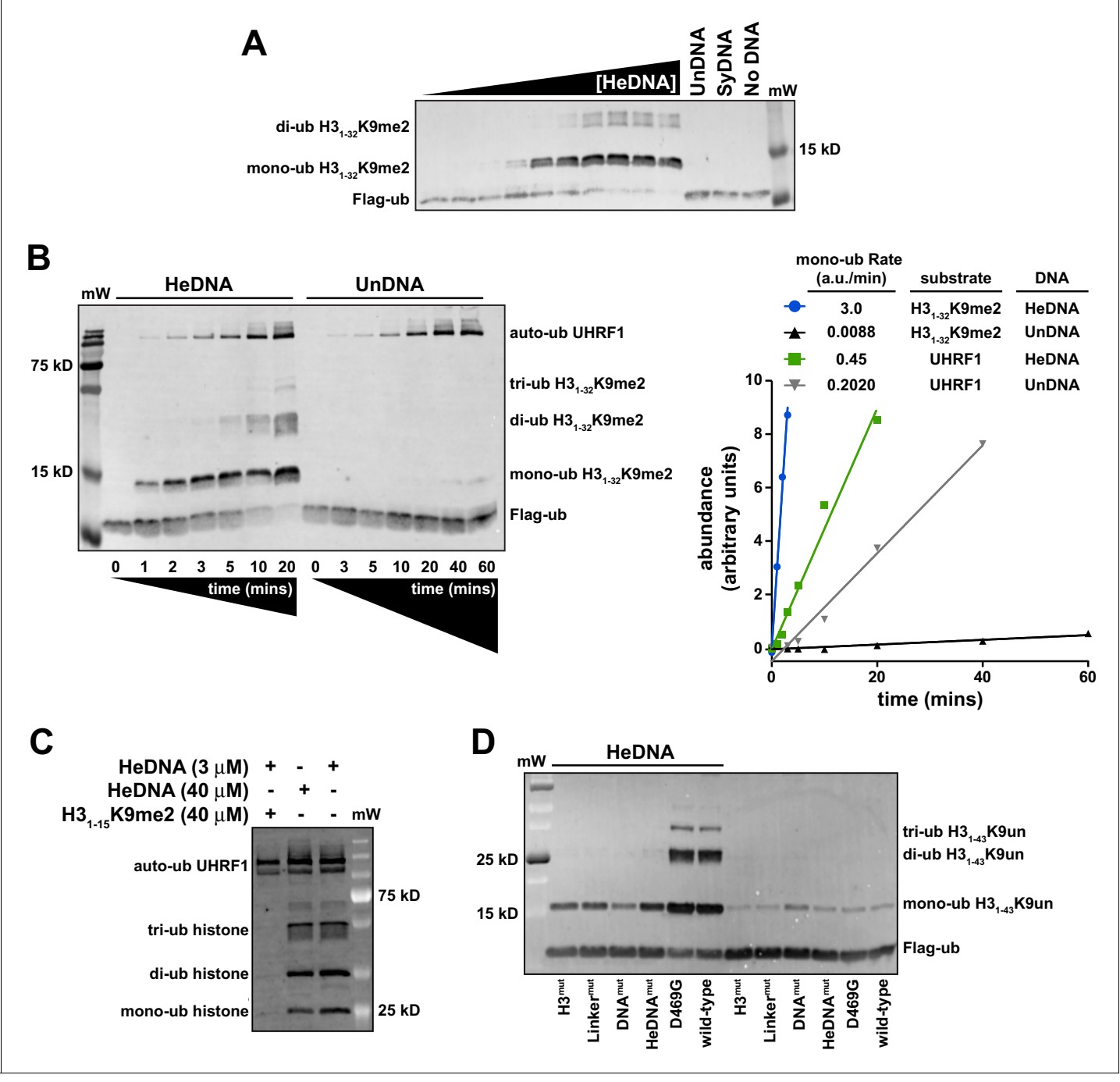

**Figure 4.** UHRF1-mediated histone H3 ubiquitylation is stimulated by substrate and HeDNA recognition. (**A**) UHRF1 ubiquitylation assays on an H3$_{1-32}$K9me2 peptide in the absence or presence of the indicated DNA oligonucleotides: HeDNA was titrated at semi-log intervals spanning 30 μM to 1 nM. SyDNA or UnDNA was added at 30 μM or 100 μM, respectively. (**B**) Rate measurement quantifying UHRF1 auto-ubiquitylation and H3$_{1-32}$K9me2 ubiquitylation in the presence of HeDNA or UnDNA at the indicated time points. Rate experiments were performed three times with similar results, and a representative blot is depicted. Blots were quantified using ImageQuant TL (GE Lifesciences). Quantified data was best described by a linear fit over the measured time scale, with the exception of HeDNA-stimulated H3$_{1-32}$K9me2 mono-ubiuitylation, which remained linear within the first 5 min of the reaction. (**C**) UHRF1 ubiquitylation assays on HeLa mononucleosomes in the presence of the indicated concentrations of HeDNA and/or an H3$_{1-15}$K9me2 peptide. (**D**) Ubiquitylation of an H3$_{1-43}$K9un peptide by UHRF1 and the indicated mutants (see *Figure 1A* for mutant annotation) in the absence or presence of HeDNA.

The following figure supplement is available for figure 4:

*Figure 4 continued on next page*

*Figure 4 continued*

**Figure supplement 1.** UHRF1 ubiquitin ligase assays.

demonstration that a DNA-protein interaction, and in particular an epigenetic modification, directly regulates enzymatic activity of an E3 ubiquitin ligase.

To further characterize HeDNA-stimulated UHRF1 ubiquitylation, we compared the rate of UHRF1 enzymatic activity on itself (auto-ubiquitylation measurements are often used as a proxy to monitor E3 ligase activity) and an H3$_{1-32}$K9me2 peptide substrate. We measured a 2.5-fold rate enhancement of UHRF1 auto-ubiquitylation in the presence of HeDNA vs. UnDNA, both in the presence or absence of H3$_{1-32}$K9me2 peptide (*Figure 4B* and *Figure 4—figure supplement 1A*). In sharp contrast, the rate of HeDNA-stimulated H3$_{1-32}$K9me2 mono-ubiquitylation was stimulated by more than 100-fold over the rate obtained with UnDNA (*Figure 4B*). Comparing the rate of activity on UHRF1 substrates (peptide vs. self), the rate of auto-ubiquitylation was 20-fold faster than the rate of peptide ubiquitylation in the presence of UnDNA. Conversely, the rate of peptide ubiquitylation was seven-fold faster than the rate of auto-ubiquitylation in the presence of HeDNA (*Figure 4B*). Based on these observations, we propose that HeDNA-binding acts as an allosteric switch to enhance ubiquitylation of histone substrates.

Similar to peptide substrates, UHRF1 mono-, di-, and tri-ubiquitylation of purified HeLa mononucleosomes was stimulated by HeDNA (*Figure 4—figure supplement 1B*), confirming that the enhanced ubiquitylation activity of UHRF1 is relevant in the context of chromatin. In addition, when excess H3$_{1-15}$K9me2 peptide (*Supplementary file 1*) (which harbors the TTD-PHD binding site but not the published ubiquitin target lysines) was added to UHRF1 mononucleosome ubiquitylation assays, H3$_{1-15}$K9me2 effectively inhibited enzymatic activity towards mononucleosome substrates (*Figure 4C*). In contrast, HeDNA concentrations as high as 40 μM did not block mononucleosome ubiquitylation, consistent with its role as an activator of UHRF1 E3 ligase activity. These results suggest that the N-terminus of H3 is the primary binding site for substrate recognition through the TTD-PHD and that DNA interaction can occur in *trans* to the nucleosome being targeted for ubiquitylation.

To further investigate the role of UHRF1 reader domain functions in ligase activity, we tested the previously described H3$^{mut}$, Linker$^{mut}$, DNA$^{mut}$, HeDNA$^{mut}$ (*Figure 1A*) and D469G (*Avvakumov et al., 2008*) mutants in ubiquitylation assays using H3$_{1-32}$K9me2 (*Figure 4—figure supplement 1C*) and H3$_{1-43}$K9un peptides as substrate (*Figure 4D*). Reacting UHRF1 with peptide substrates, we observed low ubiquitin ligase activity in the absence of HeDNA, while HeDNA binding permitted robust formation of mono-, di-, and tri-ubiquitylated H3 peptides (*Figure 4D* and *Figure 4—figure supplement 1C*). Characterizing DNA-, HeDNA-, and histone-binding loss-of-function UHRF1 mutants in ubiquitylation assays revealed defects in HeDNA-dependent H3 ubiquitylation, with the exception of the previously reported SRA loss-of-function mutant (D469G) (*Avvakumov et al., 2008*), that exhibited wild-type binding to HeDNA in our assays (*Figure 4—figure supplement 1D*). Ubiquitylation defects observed for H3$^{mut}$ and Linker$^{mut}$ confirmed a critical role for the TTD-PHD as the substrate-binding domain for HeDNA-dependent H3 ubiquitylation and demonstrated that multivalent *cis* engagement of H3K9me3 by the UHRF1 TTD-PHD is required for proper ubiquitylation (*Figure 4D* and *Figure 4—figure supplement 1C*). In addition, complete loss of HeDNA-dependent ubiquitylation for DNA$^{mut}$ UHRF1 and the absence of multi-ubiquitylated H3 for HeDNA$^{mut}$ UHRF1 further support the role of DNA binding and HeDNA recognition to fully activate UHRF1 ubiquitin ligase activity. The histone ubiquitylation defects observed for these loss-of-function mutants further highlights the interplay between UHRF1 functional domains to support proper UHRF1 ubiquitin ligase activity.

There is a growing appreciation for the role of allosteric regulation of RING E3 ubiquitin ligase activity in the field of ubiquitin biology (*Vittal et al., 2015*). Most often, the regulation of RING E3 ligases is accomplished through modulation of the E3 affinity for an E2-ub (thioesterified E2-ubiquitin) conjugate. Auto-inhibition release is the primary mechanism observed to date, in which steric occlusion of the RING domain prevents E3 interaction with the E2-ub until the E3 receives an appropriate release signal. Examples of RING auto-inhibition release include neddylation of Cullins

(*Duda et al., 2008*; *Saha and Deshaies, 2008*), phosphorylation of Cbl (*Dou et al., 2012*), and substrate/peptide mimetic binding to inhibitor of apoptosis 1 (*Dueber et al., 2011*). Taking into consideration the above-described allosteric regulatory mechanism of E3 ligase activity, we first sought to determine whether the interaction with HeDNA could affect association with E2-ub. Using isothermal titration calorimetry (ITC) and nuclear magnetic resonance (NMR) spectroscopy, we monitored the interaction between UHRF1 and E2-N-ub (isopeptide-linked C85K E2-ubiquitin conjugate) or $^{15}$N-E2-O-ub (oxyesterified C85S E2-ubiquitin conjugate), respectively. Surprisingly, neither ITC nor NMR spectrum intensity loss measurements indicated a change in affinity for the E2 conjugates in the absence or presence of HeDNA (*Figure 5A–B* and *Figure 5—figure supplement 1*). In addition, we readily observed UHRF1 auto-ubiquitylation in the presence of HeDNA and UnDNA (*Figure 4B* and *Figure 4—figure supplement 1A*), indicating the RING domain of UHRF1 could productively interact with E2-ub regardless of the methylation status of the bound DNA. Notably, upon E3 binding to the conjugated E2, NMR resonances belonging to the conjugated ubiquitin did not suffer as great a loss in intensity (*Figure 5B* and *Figure 5—figure supplement 1B–C*), indicating that ubiquitin retained its dynamics when bound to the E3. This observation suggests that UHRF1 binding does not promote closed E2-ub states as strongly as other canonical RING domain E3's (*Christensen et al., 2007*; *Pruneda et al., 2011a*, *2012*).

Recently, another mechanism of allosteric regulation of RING activity has been described where a ligand, Poly-ADP-ribose (PAR), induces a conformational change directly in the E3 RNF146 RING domain. This alternative RING conformation stabilizes the E2-ub/RING complex, thereby enhancing ubiquitin discharge from the conjugated E2 (*DaRosa et al., 2015*). To test whether HeDNA-induced UHRF1 ubiquitin ligase activity enhanced ubiquitin discharge from the conjugated E2, we performed single turnover ubiquitylation assays where purified E2-ub served as the ubiquitin donor and excess free lysine was present as a proxy ubiquitin substrate. The rate of ubiquitin discharge from E2 in these assays (monitored by the loss of the E2-ub and the appearance of free E2) showed only a modest increase in the reactivity of the conjugate in the presence of HeDNA (*Figure 5C*). These results suggest that HeDNA-dependent activation of UHRF1 RING activity does not occur through enhancement of the intrinsic rate of ubiquitin discharge from the E2 to non-specific lysine sidechains.

Remarkably, when an H3$_{1-20}$ peptide (*Supplementary file 1*) was added to ubiquitin discharge reactions, we observed rapid conversion of E2-ub to E2 (*Figure 5C*). Additionally, we observed the appearance of a band corresponding to ubiquitylated H3$_{1-20}$ (*Figure 5C*). Notably, we also observed a decrease in the amount of free ubiquitin and UHRF1 auto-ubiquitylation formed when peptide was present, presumably because more ubiquitin was being transferred to H3 (*Figure 5C*). Thus, even under conditions where free lysine was in great excess, ubiquitin was transferred rapidly and preferentially to H3 substrate in the presence of HeDNA.

To determine whether activation of UHRF1 occurs upon substrate binding (i.e., substrate-assisted activation), we performed single turnover assays in the presence of H3$_{1-20}$, H3$_{1-20}$K14acK18ac, and H3$_{1-20}$K9acK14acK18ac (*Figure 5D*, see also *Supplementary file 1*). We previously demonstrated the interaction of the UHRF1 TTD-PHD with these potential substrates by peptide microarray (*Rothbart et al., 2013*) and reasoned that the acetylated peptides would maintain interaction with the TTD-PHD but would be unable to accept ubiquitin. Neither the H3$_{1-20}$K14acK18ac nor H3$_{1-20}$K9acK14acK18ac were capable of being modified with ubiquitin (*Figure 5D*). Additionally, rapid E2-ub depletion was only observed in the sample containing H3$_{1-20}$ (*Figure 5D*), indicating that substrate binding alone does not enhance the E3 ligase activity of UHRF1 in the presence of HeDNA. Consistent with our previous single turnover results (*Figure 5C*), reactions that contained unblocked lysines on H3$_{1-20}$ accumulated less auto-ubiquitylated UHRF1 and free ubiquitin compared to assays with no peptide or with the acetylated H3 peptides (*Figure 5D*), supporting a model where HeDNA alters the substrate specificity of UHRF1 ubiquitylation. It is worth noting that in all single and multiple turnover assays performed, virtually no ubiquitylated species of H3 were observed in assays that lacked HeDNA. Taken together, these data demonstrate that HeDNA stimulates UHRF1 ubiquitin ligase activity through a novel regulatory mechanism and suggest that HeDNA binding serves as an allosteric switch that directs the TTD-PHD bound H3 substrate to the E2-ub active site for transfer.

We next used high-resolution mass spectrometry to further characterize the histone lysine specificity of HeDNA-stimulated UHRF1 ubiquitylation. HeLa mononucleosomes were reacted with UHRF1 in the presence of HeDNA or UnDNA for 2 hr using the enrichment strategy depicted in *Figure 6A*. Since histone proteins are highly basic, propionic anhydride was used to chemically

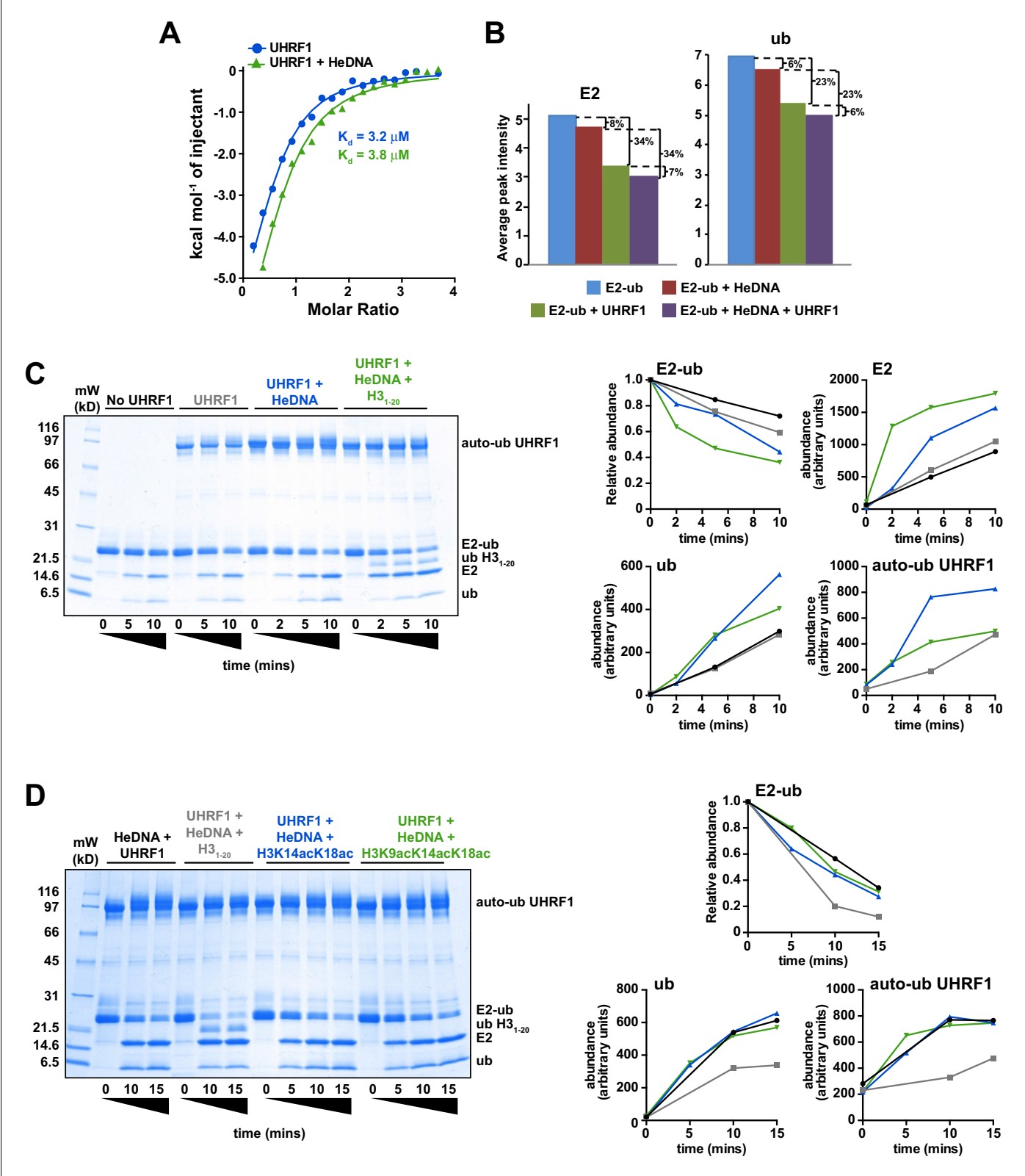

**Figure 5.** HeDNA binding directs ubiquitin to histone substrates. (**A**) ITC measuring the interaction of UHRF1 with E2-N-ub (UbcH5c(C85K)-ub linked by isopeptide bond) in the presence and absence of HeDNA (see also *Figure 5—figure supplement 1A*). (**B**) Average peak intensities for ${}^1$H-${}^{15}$N HSQC-TROSY spectra of the ${}^{15}$N-E2-o-ub (E2-o-Ub, UbcH5c(S22R/C85S)-ub esterified conjugate) (see also *Figure 5—figure supplement 1B–C*). Percentages indicate the reduction in intensity due to addition of UHRF1 or HeDNA. The addition of HeDNA to E2-ub (comparing blue to red) results in the same

*Figure 5 continued*

decrease in intensity as the addition of HeDNA to a sample containing E2-ub and UHRF1 (comparing green to purple) for both the E2 or ub within the conjugate. (C) Coomassie-stained gel of ubiquitin discharge assays in the presence of the indicated ligands and 20 mM free lysine (left). Densitometry analysis of the indicated components of the reaction (right). Line coloring corresponds to lane labels at the top of the gel. (D) Coomassie-stained gel of ubiquitin discharge assays in the presence HeDNA and either no peptide, H3$_{1-20}$, H3$_{1-20}$K14aK18ac, H3$_{1-20}$K9acK14acK18ac and 20 mM free lysine (left). Densitometry analysis of the indicated components of the reaction (right). Line coloring corresponds to lane labels at the top of the gel. We conducted at least five ubiquitin discharge assays, and the trends observed for each condition in panels C and D were consistent across all experiments.

The following figure supplement is available for figure 5:

**Figure supplement 1.** HeDNA binding does not modulate the interaction of UHRF1 with E2-ubiquitin conjugate.

modify free lysines and facilitate the identification of peptide fragments (*Garcia et al., 2007*). We first identified peptides that were enriched in the HeDNA sample relative to the UnDNA sample, and the only histone peptides that were enriched greater than ten-fold were derived from H3 (*Supplementary file 2*) (see Materials and methods for information about normalization and quantification). Consistent with our findings that HeDNA stimulates UHRF1 ubiquitin ligase activity, ubiquitin remnants on H3 peptides were heavily enriched in samples reacted in the presence of HeDNA relative to UnDNA (*Figure 6B* and *Figure 6—figure supplement 1,2*). In addition to the previously identified sites H3K18 and H3K23, we also found H3K14ub, H3K27ub, and H3K36ub heavily enriched in the presence of HeDNA (*Figure 6B*, *Figure 6—figure supplement 1,2*). We further identified abundant H3K18ub and H3K23ub (multi-ubiquitylation) on the same peptide. These results are consistent with laddering observed for H3ub in our experiments with synthetic peptides (see *Figure 4B*) and recombinant and native mononucleosomes (see *Figure 4C* and *Figure 4—figure supplement 1C*), as well as immunoblots from HeLa cells in previous studies (*Nishiyama et al., 2013*; *Qin et al., 2015*). H3K18ub was the most abundant ubiquitylated peptide based upon spectral counts (greater than 15-fold more than any other site), and H3K23ub was only observed in the context of H3K18ub (*Figure 6B*). Collectively these results suggest that H3K18 is the preferred ubiquitylation site for UHRF1, but that UHRF1 can target a number of lysines on the H3 tail.

Consistent with our model of HeDNA altering UHRF1 substrate preferences, we also observed changes to the sites of UHRF1 auto-ubiquitylation in the presence of HeDNA (*Figure 6—figure supplement 2C–E*). In particular, we identified a nine-fold enrichment of UHRF1 K303ub, a solvent exposed lysine in the PHD near the C-terminus of a bound H3 peptide (*Figure 6—figure supplement 2D*). This region may represent the target zone for HeDNA-dependent ubiquitylation where the RING domain would be in proximity to this region. Accordingly, we observed an additional auto-ubiquitylated UHRF1 band in the presence of HeDNA compared to that observed with UnDNA (*Figure 6—figure supplement 2E*), corroborating our mass spectrometry results.

We also identified several histone PTMs that co-occurred with ubiquitylated H3 peptides, including all three states of H3K9 methylation (*Figure 6B*). Additionally, several of the most enriched H3 peptides not containing ubiquitin remnants also contained H3K9me2 (*Supplementary file 2*). We confirmed this epigenetic link by immunoblotting HeLa mononucleosomes ubiquitylated by UHRF1. Ubiquitylated H3 was detected in the presence of HeDNA (but not UnDNA) on nucleosomes marked with H3K9me3, but not on nucleosomes marked with H3K9acK14ac (*Figure 6C*). In addition, titrating recombinant human histones H3, H2A, and H2B into ubiquitylation assays revealed that while UHRF1 could modify H2A and H2B at concentrations above 5 µM, H3 could be modified at submicromolar concentrations, and H3K9me2 protein (synthesized by native chemical ligation) could be modified at even lower concentrations (*Figure 6—figure supplement 3*). Taken together, these findings strongly support the role of H3K9 methylation in directing UHRF1 ubiquitylation to adjacent lysine residues in the presence of HeDNA. Other histone PTMs co-occurring on ubiquitylated peptides were: H3K23ac, also identified in another study (*Qin et al., 2015*); H3K27me2, which often co-occurs with H3K9me2/me3 and is considered a hallmark of facultative heterochromatin (*Boros et al., 2014*); H3K36ac, and H3K37me3 (*Figure 6B* and *Figure 6—figure supplement 1,2*). However, future studies will be required to dissect the biological significance of these PTM combinations to ubiquitin ligase-dependent UHRF1 function.

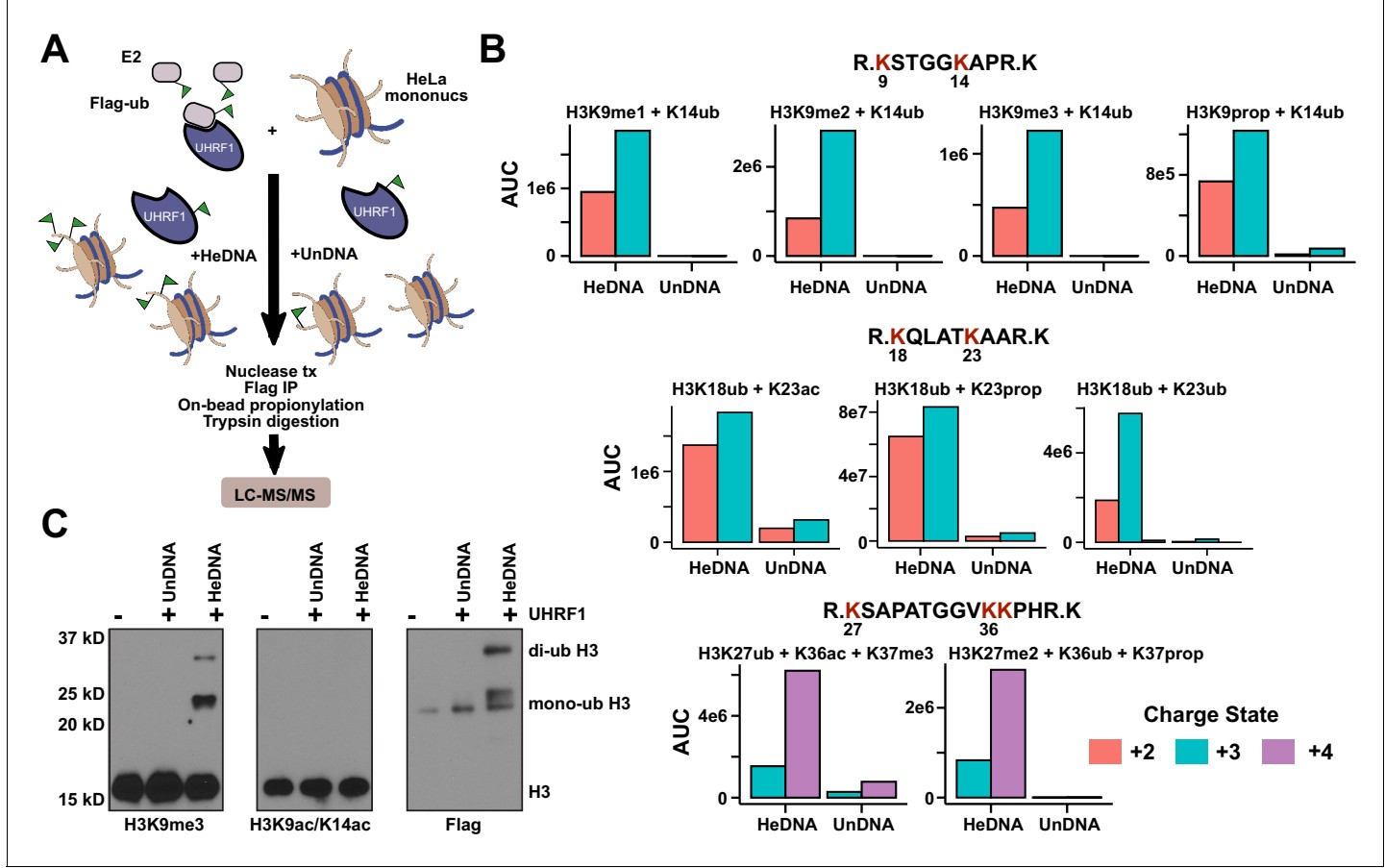

**Figure 6.** HeDNA stimulates UHRF1-directed ubiquitylation of multiple N-terminal lysines on histone H3. (A) Schematic of the assay and sample preparation strategy to identify by LC-MC/MS products of UHRF1 ubiquitylation reactions with HeLa mononucleosomes in the presence of UnDNA or HeDNA. (B) Quantification of the area under the curve (AUC) from extracted-ion chromatograms for the indicated ubiquitylated H3 peptides enriched by immunoprecipitation of FLAG-ub. See *Figure 6—figure supplement 1* for retention times and fragmentation for identified peptides. (C) Immunoblot analysis for Flag-ub and the indicated histone PTMs following UHRF1 ubiquitylation of HeLa mononucleosomes reacted in the presence of HeDNA or UnDNA, (-) indicates unreacted nucleosomes.

The following figure supplements are available for figure 6:

**Figure supplement 1.** Ion-extracted chromatograms (left) and fragmentation patterns (right) for each ubiquitylated peptide identified using the search procedures described in Materials and methods.

**Figure supplement 2.** Characterizing lysine prioritization of UHRF1 ubiquitylation on mononucleosomes.

**Figure supplement 3.** UHRF1 targets H3K9me2 histones for ubiquitylation.

## Discussion

Our studies define an orchestrated sequence of histone- and DNA-binding events targeting UHRF1 to chromatin and identify a key regulatory mechanism controlling DNA methylation inheritance through UHRF1 E3 ligase activation following recognition of HeDNA. This mechanism is consistent with the observation that UHRF1-dependent H3 ubiquitylation accumulates in S-phase when HeDNA intermediates are generated behind replicating DNA polymerase (*Nishiyama et al., 2013*; *Qin et al., 2015*). Building on recent studies connecting H3 ubiquitylation to DNMT1 recruitment (*Nishiyama et al., 2013*; *Qin et al., 2015*), we propose a model where UHRF1 is targeted to chromatin through its coordinated histone and DNA reading activities (*Figure 7A*). When UHRF1

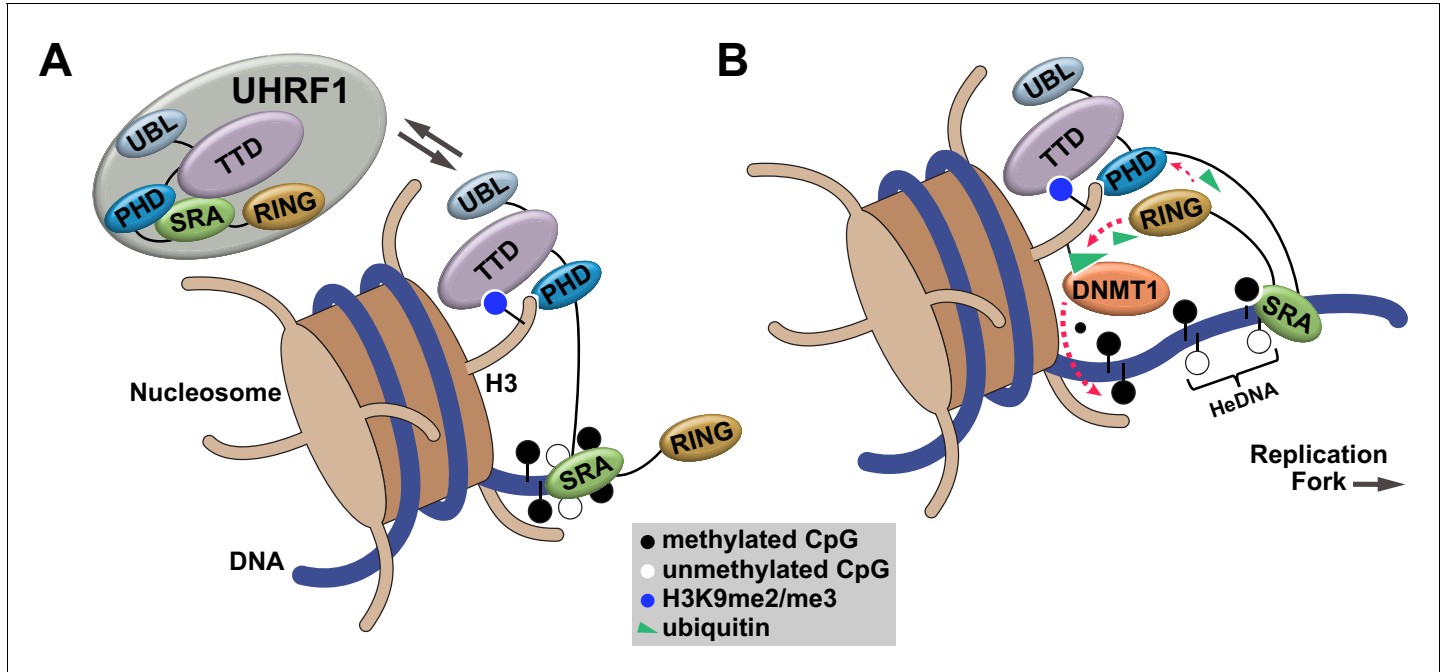

**Figure 7.** Proposed model for the contributions of DNA and histone recognition events to the DNA methylation regulatory function of UHRF1. (**A**) UHRF1 is targeted to and retained on chromatin by the combined actions of H3K9me2/me3 recognition through the TTD-PHD and DNA interaction, independent of methylation status, through the SRA. (**B**) The interaction of the SRA with HeDNA, a DNA replication intermediate, directs the ubiquitin ligase activity of UHRF1 towards N-terminal lysines on the histone H3 tail. H3 ubiquitylation by UHRF1 contributes to the retention of DNMT1 in chromatin environments enriched for HeDNA and facilitates the epigenetic inheritance of DNA methylation patterns.

The following figure supplement is available for figure 7:

**Figure supplement 1.** Mouse UHRF1 (Np95) SRA adopts different conformations bound to UnDNA (left;PDB:2ZO2) and HeDNA (right;PDB:3F8I).

encounters HeDNA, H3 ubiquitylation serves as a mechanism to facilitate the recruitment of DNMT1 to replicating regions of the genome to copy parental DNA methylation patterns (*Figure 7B*).

Structural characterization of the UHRF1 SRA bound to HeDNA (*Arita et al., 2008*; *Avvakumov et al., 2008*; *Hashimoto et al., 2008*) and cellular localization of UHRF1 with DNMT1 and PCNA (proliferating cell nuclear antigen) at replicating heterochromatic foci (*Bostick et al., 2007*; *Sharif et al., 2007*) contribute to the model in which UHRF1 ubiquitylation of S-phase chromatin is mediated through HeDNA recognition. While it remains to be seen whether uncoupling UHRF1 from HeDNA recognition changes its residence genome-wide, our studies show HeDNA sensing is not required to target UHRF1 to bulk chromatin. Rather, coordinated recognition of H3 and DNA, independent of HeDNA discrimination, drives chromatin interaction. We propose that the avidity resulting from sub-μM affinities of UHRF1 for both DNA and H3 peptides through reciprocal positive allostery provides a biochemical basis by which UHRF1 is exclusively localized on chromatin. This may also explain why small perturbations to histone binding affinity through the TTD-PHD (e.g., Linker[mut] and TTD aromatic cage mutation [*Rothbart et al., 2013*, *2012*]) so dramatically affect chromatin targeting of this protein.

Our studies define the UHRF1 TTD-PHD as the substrate-binding domain for HeDNA-stimulated ubiquitylation, further demonstrating a functional role for the coordinated recognition of H3K9me2/me3 and HeDNA in UHRF1 ubiquitylation. UHRF1 appears to be versatile in targeting lysines for ubiquitylation on the H3 tail (*Figure 6B*): this ability may be related to the complexities of PTM patterning found on this region of the H3 tail (*Young et al., 2009*) and the necessity to promote efficient recruitment of DNMT1 to differentially modified chromatin environments. In addition, UHRF1 histone ubiquitylation may serve other roles in DNA related processes (i.e., DNA repair) (*Liang et al., 2015*; *Tian et al., 2015*; *Zhang et al., 2016*). Further studies are necessary to examine

the biological consequence of different patterns of H3 ubiquitylation by UHRF1 and their relationship to pre-existing histone PTM signatures.

Important to note are studies on the enzymology of DNMT1 activity that show the enzyme has an intrinsic preference for HeDNA substrates (*Goyal et al., 2006*) and methylates in a processive manner (*Bestor and Ingram, 1983*; *Hermann et al., 2004*). Our studies define a major function for HeDNA, beyond direct stimulation of DNMT1 activity, in the regulation of UHRF1 histone ubiquitylation and DNA methylation inheritance. Considering DNMT1 behavior on oligonucleotide substrates, it is intriguing to speculate that UHRF1 functions to provide a nucleation event for DNMT1 recruitment to chromatin. Future studies mapping the genome-wide distribution of UHRF1-directed H3 ubiquitylation in relation to DNA methylation patterning will clarify the relationship between UHRF1 and DNMT1 activities.

How might HeDNA binding alter the substrate preference of UHRF1 directed ubiquitylation? We speculate that HeDNA is bound by the UHRF1 SRA in a manner that positions the RING in proximity to the H3 binding region of UHRF1. RING activity towards H3 may be conformationally restricted when UHRF1 binds SyDNA or UnDNA (*Figure 7*). Consistent with this hypothesis, structures of the SRA domain from mouse UHRF1 (Np95) show this domain can adopt different conformations bound to UnDNA and HeDNA (*Figure 7—figure supplement 1*) (*Hashimoto et al., 2008*). Also the NKR finger of the UHRF1 SRA, which harbors the HeDNA[mut], adopts a highly ordered conformation upon HeDNA binding (*Avvakumov et al., 2008*; *Hashimoto et al., 2008*). This allows for pseudo-base pairing to the exposed guanosine nucleotide (*Figure 1—figure supplement 1*), and we hypothesize this stable finger conformation is critical for HeDNA-stimulated H3 ubiquitylation. Unfortunately, efforts to crystallize the enzymatically active conformation of UHRF1 proved unsuccessful. Determining the active conformation of UHRF1 will be an important step in further understanding the regulation imparted by HeDNA.

Why might ubiquitin be an ideal PTM to accompany a temporally controlled process like replication-coupled DNA methylation? Ubiquitin itself is a functional protein domain capable of participating a wide variety of protein-protein interactions (*Harrison et al., 2016*) and can sterically occlude surfaces, as has been proposed for H2BK120ub in the formation of a productive complex with DOT1L (*Zhou et al., 2016*). Ubiquitin modifications are also dynamic and can be rapidly removed by deubiquitylases. In fact, recent analysis of ubiquitin turnover kinetics showed that the half-life of H2BK123ub in budding yeast is approximately one minute (*Yumerefendi et al., 2016*). Thus, discovering the identity of the deubiquitylase that removes H3 ubiquitylation may provide key insight into the dynamics of DNA methylation regulation at the level of histone ubiquitylation. The initial study implicating H3 ubiquitylation in the inheritance of DNA methylation indirectly suggested that USP7 may be responsible for this function through interaction with DNMT1 (*Nishiyama et al., 2013*). This is notable, as USP7 has also been shown to interact with UHRF1 (*Zhang et al., 2015*). Additionally, recent studies have tied USP7 to DNA replication (*Lecona et al., 2016*) and the maintenance of heterochromatin (*Mungamuri et al., 2016*), providing a biological link to replication-coupled inheritance of DNA methylation. However direct evidence of USP7 catalyzed deubiquitylation of H3 is lacking.

In conclusion, our study defines the relationship between UHRF1 histone-binding, DNA-binding, and ubiquitylation activities and connects HeDNA recognition to UHRF1 enzymatic function. Additionally, we characterize HeDNA as an active epigenetic mark that allosterically regulates UHRF1 ubiquitylation towards histone H3. More broadly, these finding provide a function for epigenetic patterning associated with UHRF1 beyond protein recruitment. We speculate that epigenetic mechanisms of multivalency and allostery are more widespread and add additional layers of complexity, specificity, and connectivity to chromatin recognition, modification patterning, and genome regulation.

## Materials and methods

### UHRF1 protein production

The cDNA that encodes amino acids 1–793 of human UHRF1 (full length) was cloned into a modified pGEX vector in frame with an N-terminal 6xHis-MBP tag that can be cleaved with TEV protease. *E. coli* were grown to O.D. 0.6 and induced with 600 mM IPTG overnight at 18°C. Cells were collected

by centrifugation and resuspended in lysis buffer (50 mM Tris-HCl, pH 8.0, 300 mM NaCl, 2 mM PMSF, 1 µM Bestatin, 1 µM Pepstain A, and 10 µM Leupeptin [Thermo Fisher Scientific, Waltham, MA]), lysed with sonication on ice, and cellular debris was pelleted at 15,000 x g for 30 min. The supernatant was passed over a HisTRAP nickel column (GE Lifesciences, Pittsburg, PA), washed (50 mM Tris-HCl, pH 8.0, 1 M NaCl, and 15 mM imidazole) and eluted (25 mM HEPES, pH 7.5, 100 mM NaCl, and 250 mM imidazole). Eluted protein was concentrated to ~2 mL using a 10 kDa spin concentrator (Amicon Ultra) and further purified by size-exclusion chromatography (SEC) over a Superdex S-200 (16/600) column (GE Lifesciences) in 25 mM HEPES, 100 mM NaCl, and 1 mM DTT. Monomeric fractions were pooled and concentrated to 100–200 µM. The purified protein was either used directly or was bound to MBP resin for overnight cleavage with TEV protease purified as previously described (*Tropea et al., 2009*). Cleaved UHRF1 was less stable at higher concentrations than 6xHis-MBP-UHRF1 but behaved similarly in binding and ubiquitylation assays. To complete the study, we purified UHRF1 from bacteria more than 10 times, and all protein preparations were functional and behaved similarly. Mutations were introduced into cDNAs by Quick Change (Agilent, Santa Clara, CA) and purified mutant proteins behaved similarly to wild-type protein, but were generally less stable at higher concentration. To circumvent this issue, UHRF1 mutants were characterized as MBP fusions.

## Fluorescence polarization binding assays

Histone peptides N- and C-terminally labeled with 5-carboxyfluorescein (FAM) were synthesized as described (*Rothbart et al., 2013*). 6-FAM-labeled double-stranded DNA was generated by annealing the following combinations of synthetic oligonucleotides (Eurofins, Louisville, KY); FAM-5'-CCATGXGCTGAC-3' and 5'-GTCAGYGCATGG-3', where X and Y are both cytosine (UnDNA), X is cytosine and Y is 5mC (HeDNA), or X and Y are both 5mC (SyDNA). Binding experiments were performed in 25 µL in black flat-bottom 384-well plates (Corning, Tewskbury, MA). Protein was titrated with 10 nM FAM-labeled DNA or histone peptides in buffer containing 25 mM HEPES, pH 7.5, 0.05% NP-40, 100 mM NaCl (unless otherwise indicated). Where indicated, 10 µM unlabeled DNA or histone peptide was included in the reaction mix. Following a 10 min incubation period, fluorescence polarization measurements were performed at 25°C with a PHERAstar fluorescence microplate reader (BMG Labtech, Cary, NC) using a 480-nm excitation filter and 520/530 ± 10-nm emissions filters. Gain settings in the parallel (∥) and perpendicular (⊥) channels were calibrated to a polarization measurement of 100 milli-polarization units (mP) for the FAM tracer in the absence of protein. Polarization (P) was determined from raw intensity values of the parallel and perpendicular channels using the equation $P = \parallel - \perp / \parallel + 2(\perp)$ and converted to anisotropy (A) units using the equation $A = 2P / 3 - P$. Equilibrium dissociation constants ($K_d$) were determined by non-linear regression analysis of anisotropy curves using a one-site binding model in GraphPad Prism. To control for variability in salt concentration, each experiment included a wild-type protein as a reference. Accordingly, the methyl preference for DNA binding (HeDNA, SyDNA, and UnDNA) and the positive allostery of histone and DNA binding of the wild-type protein was observed in greater than ten independent experiments in various buffers and salt concentrations with several batches of purified protein.

## Ubiquitylation assays

Ubiquitylation assays were typically performed in 20 µL reactions containing 1.5 µM UHRF1, 100 nM E1 activating enzyme (Boston Biochem #E-304; Cambridge, MA), 200 nM E2 Ubc5c (purified in house over HisTRAP column), 2.5 mM MgCl₂, 1 mM DTT, 5 µM FLAG-ubiquitin (Boston Biochem), 10 mM ATP, 25 mM HEPES, pH 7.5, and 100 mM NaCl. Unless otherwise indicated, peptide concentrations were 13 µM, and HeDNA, SyDNA, and UnDNA concentrations were 3 µM, 10 µM, and 40 µM, respectively. Assays were performed at 25°C and quenched after 20 min with SDS-PAGE loading buffer (2% SDS, 10% glycerol, 1% 2-Mercaptoethanol, 50 mM Tris-HCl pH 6.8, 0.01% bromophenol blue). Reactions were ran on 16% SDS-PAGE gels, transferred to PVDF membranes, and visualized using fluorescent imaging of immunoblots probed for FLAG-ubiquitin with FLAG (Sigma #F3165, 1:5000; St. Louis, MO or BioLegends #637304, 1:5000; San Diego, CA) and Alexa Fluor 488 or 647 (Life Technologies 1:5,000; Carlsbad, CA) antibodies on a Typhoon Trio+ fluorescent scanner (GE Lifesciences). Histone peptide substrates were synthesized as previously described (*Rothbart et al., 2013*). Recombinant histone proteins and mononucleosomes were obtained

commercially from Epicypher (H2A, #15–0301; H2B, 15–0302; and H3.1, #15–0303; mononucleosomes, #16–0002; Research Triangle Park, NC). Allosteric activation of UHRF1 ubiquitylation activity towards histone peptides and nucleosomes was observed in more than ten independent experiments, and DNA and UHRF1 titrations were repeated three times. The activities of the mutant proteins were tested in five independent experiments with at least two protein preparations for each mutant. Rate measurements for UHRF1 ubiquitylation activities in the presence of HeDNA vs UnDNA were conducted three times with similar results as *Figure 4B*.

## Synthesis of H3K9me2 histone using native chemical ligation

C-terminal H3 peptide (amino acids 11–135; T11C) was prepared as described (*Shogren-Knaak et al., 2003*) by cleavage of precursor with Factor 10X. Purification by reverse-phase HPLC followed by pooling of appropriate fractions and lyophilization afforded a white solid (6.2 mg). The theoretical mass of $C_{622}H_{1040}N_{196}O_{172}S_3$ product is 14112.54 Da and the measured mass of the product was 14112.94 Da. N-termial peptide thioester ARTKQTARK(me2)S-Mes-OH was synthesized as described (*Mahto et al., 2011*) and purified by reverse-phase HPLC to 70% purity. After purification peptide contained 30% of hydrolysis product (ARTKQTARK(me2)S-OH). A mixture of 1 mg of C-terminal peptide (70.86 nmoles) and 0.52 mg of an N-terminal peptide thioester (70% pure; 280.3 nmoles; 4 molar equivalents) in 0.5 mL of ligation buffer (3 M Guanidine-HCl, pH 7.9, 100 mM potassium phosphate) was treated with benzyl mercaptan (2.5 µL) and thiophenol (2.5 µL), and the mixture shaken vigorously for 24 hr. The reaction mixture was diluted with ligation buffer (500 µL), treated with MeCN:water:trifluoroethanol (750 µL; 25:75:0.1), and desalted by dialysis (2 x 30 min with water change). Analysis by reverse-phase HPLC and by gel electrophoresis on SDS-18% polyacrylamide gel followed by staining with coomassie blue indicated a complete ligation reaction. Purification by reverse-phase HPLC followed by pooling of appropriate fractions and lyophilization afforded H3K9me2 (1–135) T11C as a white solid (0.70 mg; 65%). The theoretical mass of $C_{670}H_{1129}N_{215}O_{186}S_3$ is 15268.90 Da and the measured mass of the product was 15269.19 Da. Ligated peptide (H3K9me2 T11C; 0.7 mg) was dissolved in argon-degassed desulfurization buffer (200 mM phosphate, 6 M guanidine-HCl, pH 6.7; 0.15 mL) and treated with ethanethiol (2 µL), TCEP (0.15 mL of 0.5 M in desulfurization buffer), t-butanethiol (10 µL), and VA-061 (2,2'-azobis[2-(2-imidazolin-2-yl)propane]) in methanol (2 µL of 0.2 M solution) and incubated at 37°C for 24 hr. The resultant mixture was purified by reverse-phase HPLC followed by pooling of appropriate fractions and lyophilization to afford H3 1–135 T11A as a white solid (0.55 mg). The theoretical mass of $C_{670}H_{1129}N_{215}O_{186}S_2$ is 15236.84 Da and the measured mass of product was 15237.16 Da.

## Lysine discharge assays

Lysine reactivity assays were performed as previously described (*DaRosa et al., 2015*; *Wenzel et al., 2011*). Briefly, the UbcH5c-Ub conjugate was generated in 25 mM sodium phosphate, pH 7.0 and 100 mM NaCl containing 1.5 µM human E1, 250 µM Ub, 100 µM UbcH5c, 2.5 mM MgCl2, and 2 mM ATP (Sigma). Reactions were incubated for 40 min at 37°C, then purified by SEC to isolate E2-Ub. SEC-purified E2-Ub was added to UHRF1 E3 samples incubated with HeDNA or buffer for 30 min on ice to form a final concentration of 8 µM E3, 25 µM E2-Ub, and, where indicated, 13 µM HeDNA and 12 µM peptide. After a zero min time point was taken, buffered L-lysine HCl (Sigma) was added to a final concentration of 20 mM and samples were incubated at 35°C, removing samples at indicated time points. Samples were quenched in non-reducing SDS sample loading buffer and analyzed by SDS-PAGE stained with either Coomassie or Oriole fluorescent gel stain (Bio-Rad, Hercules, CA). Lysine reactivity assay performed in the presence of excess free lysine were performed in the following conditions: 32 µM E2-Ub (UbcH5c and WT Ub), 8 µM UHRF1, 13 µM HeDNA, 11 µM H3$_{(1–20)}$K9me3, with 20 mM Lysine. Ubiquitin discharge assays were performed at least five times in the lab and yielded results consistent with *Figures 5C–D*.

## Pull-down assays

His-MBP-tagged UHRF1 SRA-RING (amino acids 405–793) was produced in *E. coli* as described above. GST-tagged UHRF1 TTD-PHD (amino acids 123–366) and BPTF PHD-Bromo (gift from Dr. Alex Ruthenburg [*Ruthenburg et al., 2011*]) were produced as previously described (*Rothbart et al., 2013*). Proteins (each at 1 µM) were incubated overnight at 4°C with MBP magnetic

beads (NEB, Ipswich, MA) in binding buffer containing 50 mM Tris-HCl, pH 8.0, 100 mM NaCl, 0.1% NP-40, 0.5% BSA, and, where indicated, 25 µM DNA oligonucleotides or histone peptides. Pulldown experiments with the SRA-RING DNA$^{mut}$ were performed with 5 µM DNA. Samples were washed extensively with binding buffer, eluted in SDS sample buffer, resolved by SDS-PAGE, transferred to PVDF membrane (Thermo), and probed with GST antibody (Sigma #G7781, 1:2,000). Pull-down assays were performed in triplicate.

## Chromatin association assays

Asynchronously growing HeLa cells were harvested by trypsinization 48 hr post transfection with the indicated FLAG-tagged human UHRF1 constructs. Pellets were washed once with cold 1x PBS, snap frozen in liquid N$_2$ and either processed immediately or stored at −80℃. Cell pellets were resuspended in 1x volume CSK buffer (10 mM PIPES pH 7.0, 300 mM sucrose, 100 mM NaCl, 3 mM MgCl$_2$, 0.1% Triton X-100 and 1x Complete EDTA-Free protease inhibitor cocktail from Roche) and incubated on ice for 20 min. Total protein was quantified by Bradford Assay (BioRad), and 10% of this total fraction was combined with an equivalent volume of CSK buffer supplemented with Universal Nuclease (Thermo, 1:5,000). Note that the concentration of the total fraction is now 0.5x. The remaining cell lysate was centrifuged at 1300 x g for 5 min at 4℃. The supernatant (soluble fraction) was collected. The chromatin pellet was resuspended in 1x volume CSK buffer and kept on ice for 10 min before being spun again at 1300 x g for 5 min at 4℃. The supernatant was discarded and the chromatin pellet was solubilized in CSK buffer supplemented with Universal Nuclease. 1–5 µg of protein from each fraction (estimated from Bradford on total extract) was resolved by SDS-PAGE, transferred to PVDF membrane (Thermo), and probed with the indicated antibodies (Flag, Sigma #F1804, 1:5,000; β-tubulin, Millipore #05–661, 1:5,000, H3, Epicypher #13–0001, 1:25,000).

## DNA methylation analysis

Immunofluorescence analysis of 5mC content was performed essentially as described with the following modifications (*Rothbart et al., 2012*). HeLa cells grown in 4-well chamber slides (Nunc Lab-Tek) were fixed with ice-cold methanol at −20℃ for 10 min. To denature the DNA, fixed cells were treated with 2 N HCl for 30 min at 37℃ and washed twice with 0.1 M boric acid, pH 8.5. Cells were blocked for 30 min in PBS containing 1% (w/v) BSA and labeled with an anti-5mC antibody (Active Motif #39649, 1:500; Carlsbad, CA) in PBS containing 1% BSA for 1 hr at room temperature. Cells were washed with PBS and incubated with an Alexa Fluor 647-conjugated secondary antibody (Life Technologies #A21236, 1:1000) for 1 hr at room temperature protected from light. Cells were washed with PBS and mounted with SlowFade Gold Antifade with DAPI (Thermo #S36942). Images were acquired using a Nikon A1+ RSi confocal microscope using a 60x objective following excitation with 403-nm and 640-nm solid-state lasers. The 5mC signal from each image was quantified using the equation $\frac{\sum_i 1_{[b_i > t]} 1_{[r_i > t]} (r_i - t)}{\sum_i 1_{[b_i > t]}}$, where $b_i$ is DAPI signal intensity for an individual pixel, $r_i$ is 5mC signal intensity for an individual pixel, and $t$ defines the background signal threshold. The percent of control 5mC was calculated using the mean 5mC signal from at least four fields of view.

## Preparation of mass spec samples

20 µg of Hela extracted mononucleosomes (Epichyper #16–0002) were used as substrate in each ubiquitylation reaction supplied with either HeDNA or UnDNA (described above) for 2 hr. The reactions were placed on ice, treated with Universal Nuclease (Thermo, 1:5,000), and the ubiquitylated products were immunoprecipitated with FLAG M2 magnetic beads (Sigma). The resin was washed 3x with 1 mL of wash buffer (HEPES, pH 7.5, 100 mM NaCl), split in half, and the beads were transferred to spin columns (Vivacon) and sequencing grade modified trypsin (Promega, Madison, WI). Half of the sample was reacted with proprionic anhydride (Alfa Aesar) using a modified version of this procedure (*Lin and Garcia, 2012*). 100 µl of a 1:3 ratio of proprionic anhydride diluted in 100 mM NH$_4$CO$_3$, pH 8.0 was added to each spin column followed by 50 µl NH$_4$OH to adjust the pH to 8.0. Each reaction was incubated for 30 min at 30℃ before being spun through the column. This protocol was repeated to ensure complete proprionylation of free lysines in the sample. The proprionylated and unreacted samples were then digested on resin using sequencing grade

modified trypsin (Promega) digested at 37°C for 2 hr. This mixture was analyzed with LC-MS/MS without proprionylation of the free amines exposed after trypsin digestion.

## LC-MS-MS

The peptide mixture was analyzed in positive mode using a nanoAquity UPLC coupled LTQ Orbitrap Elite mass spectrometer (Thermo ). Chromatographic separation used a 2 cm trapping column (Acclaim PepMap 100) and a 15 cm EASY-spray analytical column (75 µm ID, C18 beads of 3.0 µm particle size, 100 Å pore size). The HPLC flow rate was set to 350 nL/min over a gradient of 1% buffer B (0.1% formic acid in acetonitrile) to 25% buffer B in 150 min. The full mass scan (300 to 2000 $m/z$) was acquired at a resolution of 120,000 with a maximum injection time of 500 ms, and MS/MS was performed in a data-dependent manner for the top 15 intense ions in the linear ion trap by collision-induced dissociation. Raw data were converted to mzXML format using ProteoWizard (*Kessner et al., 2008*) and searched using the Crux pipeline (*McIlwain et al., 2014*) (version 2.1.16867) against the human UniProtKB/Swiss-Prot sequence database (downloaded on 2/20/15) (*Boutet et al., 2007*). Search parameters were set as the following: peptides between 6 and 25 amino acids long with a precursor mass tolerance of 0.5 amu, no missed cleavages, fully-enzymatic Arg-C digestion, a static propionyl modification (+56.026215) on lysines, and a maximum of 4 variable modifications consisting of up to 2 lysine ubiquitinations (+58.016716), 2 methylations (+14.01565), 2 dimethylations (−27.994915), 2 trimethylations (−13.979264), 2 acetylations (−14.015644), 1 methionine oxidation (+15.99492), and 1 STY phosphorylation (+79.966331). The mass of propionyl was subtracted from variable lysine modification masses (except methylation) due to the already applied static propionyl modification. For unpropionylated samples, the differing parameters were: up to 3 missed cleavages, fully-enzymatic trypsin digestion, no static modifications, and a maximum of 4 variable modifications consisting of up to 2 lysine ubiquitinations (+114.042931), 2 methylations (+14.01565), 2 dimethylations (+28.0313), 2 trimethylations (+42.046951), 2 acetylations (+42.010571), 1 methionine oxidation (+15.99492), and 1 STY phosphorylation (+79.966331). Prior to execution of the Percolator algorithm supplied by Crux, deltaCn scores were re-computed using an alternate definition: $deltaCn_i = 1 − ((xcorr_1-xcorr_i) / xcorr_1)$. This adjustment was performed because the similar mass of trimethylation and acetylation results in identical xcorr values for the low mass accuracy MS/MS spectra from linear ion traps, which then led to invalid deltaCn values with the default equation used by Percolator. After application of a 5% FDR threshold, peptides were further filtered by ensuring they had the expected retention time relative to peptides having the identical unmodified sequence. We used the following procedure. First, peptides with the same unmodified sequence were sorted in ascending order by their Percolator PEP (posterior error probability). Then, each peptide (starting from lowest to highest PEP) was accepted if at least one of its MS/MS scan's retention time was consistent relative to all currently accepted peptides having the same unmodified sequence. The expected relative retention time constraints were: ubiquitin < dimethyl ≤ trimethyl < acetyl < propionyl < methyl, oxidation < unmodified, and phosphorylation ≤ unmodified. Peptides expected to have the same retention times were allowed to elute within 2 min of each other. Finally, peptide H3K9me3 + K14ub was accepted after manual inspection of its corresponding MS/MS spectra, isotopic distribution, and its consistent retention time despite being above the 5% FDR threshold. Quantification was performed within Skyline (*MacLean et al., 2010*) and the results were exported for further visualization and analysis using the R programming language. Proteomics data have been deposited to the ProteomeXchange Consortium via the PRIDE partner repository with the dataset identifier PXD003983.

## Analytical size exclusion

A 10 µM solution of apo-UHRF1 or 1:1:1 ratio of ligands (HeDNA and H3$_{(1–15)}$K9me3 peptide) was passed over a Superdex 200 (10/300) GL column using an AKTA purifier FPLC (GE Lifesciences) in size exclusion buffer (25 mM HEPES pH 7.4, 100 mM NaCl, 1 mM DTT) with a flow rate of 0.5 mL/min. Samples with ligand were allowed to equilibrate for 10 min prior to injection onto the column. The apparent molecular weight was calculated using a linear fit to the retention time for a set of molecular weight standards (BioRad #1511901). Analytical size exclusion experiments were repeated three times with identical results.

## Dynamic light scattering

Dynamic light scattering was measured using were a DynaPro Plate Reader (Wyatt Technology, Goleta, CA ). UHRF1 was at 5 μM and 10 μM H3$_{(1-15)}$K9me2 or DNA was added to a final volume of 50 μL in buffer (HEPES pH 7.5 100 mM NaCl and 1 mM DTT). Samples were incubated for 10 min before monitoring light scattering for over 100 s for each sample. Light scatter for each ligand alone yielded a low intensity and poly-dispersed signal that did not significantly contribute to the scattering when UHRF1 was present. Addition of the ligand however likely accounts for the small increases to poly-dispersity observed upon addition of ligand.

## Atomic force microscopy

A 20 nM solution of UHRF1 (25 mM HEPES pH 7.4, 100 mM NaOAc, 1 mM DTT) was mixed with or without HeDNA (5 μM) and deposited on freshly peeled mica, immediately rinsed with water (Sigma #W4502), and dried with nitrogen gas before imagining. All images were acquired on the same day as the deposition. Images were collected on an MFP3D Atomic Force Microscope (Asylum Research Oxford Instruments using the following parameters: scan rate 1 Hertz, scan size 1 μM x 1 μM, image resolution 1024 x 512. Images were collected in intermittent contact mode (AC mode) using AFM probes from NanoSensor (PPP-FMR, force constant = 2.8 N/m). Images were analyzed using the Asylum Research AFM software package. The images were flattened to a second-degree polynomial to account for surface warping artifacts and volume analysis was performed using built-in particle analysis (a more detailed review of this methodology can be found here *Ratcliff and Erie, 2001*). Volume distributions were plotted to a peak fit model and visualized using Origin 6.1 (origin labs). The fact we could only identify a single volume species indicates monomeric UHRF1; the kD (data not shown) we calculated from AFM volume is also in agreement with monomeric UHRF1.

## NMR

Oxyester-linked 15N E2-O-Ub conjugate (UbcH5c(Ser22Arg/Cys85Ser)-O-Ub was generated as previously described (*Pruneda et al., 2011b*). Two-dimensional $^1$H-$^{15}$N HSQC-TROSY experiments were performed with 200 μM $^{15}$N E2-O-Ub conjugate in 25 mM sodium phosphate, pH 7.0 and 150 mM NaCl on a Bruker 500 MHz AVANCE II NMR spectrometer. MBP-UHRF1 and/or HeDNA was added to experiments to a final concentration of 18 μM and 22 μM, respectively. NMR data was processed with NMRPipe (*Delaglio et al., 1995*) and peak intensities were determined using NMRViewJ (*Johnson and Blevins, 1994*) (OneMoonScientific). Relative peak intensity changes were determined as the absolute peak intensity divided by the initial intensity of the E2-O-Ub conjugate in the absence of additives.

## Isothermal calorimetry (ITC)

E2-N-ub was generated as previously described (*Branigan et al., 2015*). ITC experiments were performed at 25°C in a MicroCal iTC200 in 25 mM Hepes pH 7.4 and 100 mM NaCl. UHRF1 was at 12 μM and the E2-N-ub was at 218 μM. Data was fit to a single site binding model with Origin.

## Acknowledgements

We thank Ashutosh Tripathy in the UNC Macromolecular Interactions Facility for assistance with ITC and DLS measurements. We also acknowledge members of the Rothbart and Kuhlman labs for helpful discussions and critical reading of the manuscript. The Structural Genomics Consortium is a registered charity (number 1097737) that receives funds from AbbVie, Bayer Pharma AG, Boehringer Ingelheim, Canada Foundation for Innovation, Eshelman Institute for Innovation, Genome Canada through Ontario Genomics Institute, Innovative Medicines Initiative (EU/EFPIA) [ULTRA-DD grant no. 115766], Janssen, Merck & Co., Novartis Pharma AG, Ontario Ministry of Economic Development and Innovation, Pfizer, São Paulo Research Foundation-FAPESP, Takeda, and the Wellcome Trust.

# Additional information

## Funding

| Funder | Grant reference number | Author |
|---|---|---|
| Van Andel Research Institute | | Scott B Rothbart |
| National Institutes of Health | T32-CA009156 | Joseph S Harrison |
| National Institutes of Health | CA181343 | Scott B Rothbart |
| National Institutes of Health | GM073960 | Brian Kuhlman |
| National Institutes of Health | GM110058 | Brian D Strahl |
| National Institutes of Health | GM088055 | Rachel E Klevit |
| National Institutes of Health | T32-GM07270 | Paul A DaRosa |

The funders had no role in study design, data collection and interpretation, or the decision to submit the work for publication.

## Author contributions

JSH, PAD, KK, SBR, Conception and design, Acquisition of data, Analysis and interpretation of data, Drafting or revising the article; EMC, DG, Conception and design, Acquisition of data, Analysis and interpretation of data; ZML, PJB, Acquisition of data, Analysis and interpretation of data; FY, Conception and design, Acquisition of data; BMD, CHA, MBM, Conception and design, Analysis and interpretation of data; AHG, DVC, LK, Acquisition of data; DAE, Analysis and interpretation of data; REK, Conception and design, Drafting or revising the article; BK, BDS, Conception and design, Analysis and interpretation of data, Drafting or revising the article

## Author ORCIDs

Krzysztof Krajewski, http://orcid.org/0000-0001-7159-617X
Scott B Rothbart, http://orcid.org/0000-0003-1631-5392

# Additional files

## Supplementary files

• Supplementary file 1. Table of synthetic peptides used in this study.

• Supplementary file 2. Table of peptides enriched greater than 10-fold in HeDNA sample relative to UnDNA sample.

## Major datasets

The following dataset was generated:

| Author(s) | Year | Dataset title | Dataset URL | Database, license, and accessibility information |
|---|---|---|---|---|
| Joseph S Harrison, Dennis Goldfarb, Michael B Major, Brian Kuhlman, Scott B Rothbart | 2016 | Hemi-methylated DNA regulates DNA methylation inheritance through allosteric control of H3 ubiquitylation by UHRF1 | http://proteomecentral. proteomexchange.org/ cgi/GetDataset?ID= PXD003983 | Publicly available at the Proteome Xchange (accession no: PXD003983) |

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
