## [Decision Letter]

Thank you for submitting your article "Hemi-methylated DNA regulates DNA methylation inheritance through allosteric activation of H3 ubiquitylation by UHRF1" for consideration by *eLife*. Your article has been reviewed by three peer reviewers, including Jerry Workman who is a member of our Board of Reviewing Editors, and the evaluation has been overseen by Ivan Dikic as the Senior Editor.

The reviewers have discussed the reviews with one another and the Reviewing Editor has drafted this decision to help you prepare a revised submission.

Summary:

This is a detailed biochemical and proteomic analysis of the role of H3K9me3 and hemi-methylated DNA on the functions of the UHRF1 ubiquitin ligase. Previous studies have implicated UHRF1binding to HeDNA to be important for the recruitment of UHRF1. However, in this very systematic biochemical study the authors find that HeDNA is an allosteric activator of UHRF1 expanding its lysine targets in histone H3. The work is well done and timely. In general the chromatin field considers protein interactions with epigenetic marks to be the basis for recruitment of chromatin modifiers. This study clearly indicates allosteric activation of chromatin modifiers needs to also be considered.

Essential revisions:

The manuscript provided by Harrison et al., describes an excellent body of work – in terms of both breadth and interest – regarding the activity of a critical multi-domain epigenetic component, UHRF1. This protein has a well-established role in recognition of hemi-methylated DNA, association with H3K9me3 and subsequent recruitment of DNMT1 for methylation maintenance. In this paper they advance mechanistic understanding of UHRF1 activity and promote a model whereby hemi-methylated DNA (HeDNA) actively promotes UHRF1-mediated ubiquitylation of H3. First they show that while HeDNA discrimination mutants of UHRF1 are unable to sustain methylation in vivo, they are able to associated with chromatin. Next they provide evidence for positive allostery, showing that UHRF1 association with H3 is promoted by the presence of (He)DNA and visa versa. They provide a mechanism for this positive allostery, showing that the SRA and TTD-PHD domains normally associate with each other but that his interaction is abrogated in the presence of DNA, which presumably allows the free TTD-PHD domain to associate with its known chromatin target, H3K9me3. Next they provide compelling evidence that the presence HeDNA stimulates the ubiquitylation activity of UHRF1 on H3 substrate. To explain the stimulatory effect of HeDNA, they aim to provide evidence for 'lysine prioritization' (weakest part of the manuscript, see below). Finally, they use mass spectroscopy techniques to determine the PTMs associated with HeDNA simulated UHRF1 activity on nucleosomes.

While this article is slightly foreshadowed by the article of Fang et al., (published earlier in April http://www.ncbi.nlm.nih.gov/pubmed/27045799), I believe the article herein still contains more than sufficient novel insight – particularly regarding ubiquitylation – to justify publication in a prestigious journal such as *eLife*. However, I felt that the authors were overly dismissive of the results from the other group, despite their findings being largely complementary. In fact, some of the data presented here could be benefit from interpretation in context of the work of Fang et al. The main concern I have with this manuscript regards their lysine prioritization model, for which I believe their current supporting evidence to be insufficient (see below).

Major concern:

1) In the Results section, the authors state: "UHRF1-dependent histone ubiquitylation was largely unaffected by free lysine in multi-turnover ubiquitylation assays, while UHRF1 auto-ubiquitylation was reduced (Figure 5)."

Looking at Figure 5 am not convinced that the presence of free lysine preferentially affects auto-ub over H3 peptide. If at all, auto-ub levels appear barely increased in the absence of free lysine, however, the di and mono-ub levels of H3 seem equally if not even more increased (cf. lanes 1 and 4 at di-ub or lanes 2 and 5 at mono-ub) suggesting that there is no prioritization of H3 lysines whatsoever. Nor is there any attempt to quantify this result.

They go on to say "We also confirmed this result using single turnover lysine discharge assays (Figure 5Figure 5—figure supplement 1)…" – yet these assays are in no way a confirmation of prioritization, as they simply show that H3 peptide ub can occur in the presence of free lysine, however, critically, they do directly compare rates H3 ub rates (normalized to auto-ub) in the absence vs. presence of free lysine as in Figure 5.

To me these results are much more consistent with the findings from Fang et al., and also shown here (Figure 3) whereby the combination of HeDNA, H3K9me3 and H3R2 unlock UHRF1 in an open confirmation associated with H3, facilitating efficient ubiquitylation of histone lysines. This more parsimonious possibility is even alluded to by the authors: Fifth paragraph of Discussion section – "We also cannot rule out the possibility that the increased rate of HeDNA-dependent ubiquitin transfer to lysines compared to auto-ubiquitylation or ubiquitin discharge to free lysine may arise from the UHRF1-bound histone lysines possessing ideal characteristics for accepting ubiquitin (e.g. spatially pre-arranged, protected from solvent, and/or appropriate pK)."

Overall, unless considerably improved evidence supporting the 'lysine prioritization' model is provided, this line of argument should be removed from the manuscript.

2) In my view the most important aspect of this paper is to establish the mechanism through which HeDNA binding stimulates ubiquitylation. This maybe partly explained by DNA binding disrupting the SRA-RING interaction with TTD-PHD (leading to enhanced binding to H3K9me), but this occurs with any DNA binding (modified or otherwise). So we are left, somewhat disappointingly, with the speculation that HeDNA binding causes specific conformational changes that reposition the RING domain to target the E2 ligase to particular H3 lysines.

---

## [Author Response]

*Essential revisions:*

*The manuscript provided by Harrison et al., describes an excellent body of work – in terms of both breadth and interest – regarding the activity of a critical multi-domain epigenetic component, UHRF1. This protein has a well-established role in recognition of hemi-methylated DNA, association with H3K9me3 and subsequent recruitment of DNMT1 for methylation maintenance. In this paper they advance mechanistic understanding of UHRF1 activity and promote a model whereby hemi-methylated DNA (HeDNA) actively promotes UHRF1-mediated ubiquitylation of H3. First they show that while HeDNA discrimination mutants of UHRF1 are unable to sustain methylation* in vivo*, they are able to associated with chromatin. Next they provide evidence for positive allostery, showing that UHRF1 association with H3 is promoted by the presence of (He)DNA and visa versa. They provide a mechanism for this positive allostery, showing that the SRA and TTD-PHD domains normally associate with each other but that his interaction is abrogated in the presence of DNA, which presumably allows the free TTD-PHD domain to associate with its known chromatin target, H3K9me3. Next they provide compelling evidence that the presence HeDNA stimulates the ubiquitylation activity of UHRF1 on H3 substrate. To explain the stimulatory effect of HeDNA, they aim to provide evidence for 'lysine prioritization' (weakest part of the manuscript, see below). Finally, they use mass spectroscopy techniques to determine the PTMs associated with HeDNA simulated UHRF1 activity on nucleosomes.*

We thank the editors and referees at *eLife* for careful review of our manuscript. We are pleased that the reviewers share our enthusiasm for these important findings, and we are grateful for the constructive feedback we received. We’ve thoughtfully considered and addressed each point raised by the reviewers. Below we provide a point-by-point response to the reviewer comments and describe how the revised manuscript has improved as a result of this critique.

*While this article is slightly foreshadowed by the article of Fang et al., (published earlier in April http://www.ncbi.nlm.nih.gov/pubmed/27045799), I believe the article herein still contains more than sufficient novel insight – particularly regarding ubiquitylation – to justify publication in a prestigious journal such as eLife. However, I felt that the authors were overly dismissive of the results from the other group, despite their findings being largely complementary. In fact, some of the data presented here could be benefit from interpretation in context of the work of Fang et al.*

We have reorganized and added text comparing similarities and differences between the analysis of histone- and DNA-binding activities of UHRF1 performed by our group and Fang et al. To better contrast and compare our findings, we moved our analysis of the Fang et al. experiments to the Results section of our paper immediately following comparative data. Several major points of comparison are highlighted below

While our collective data suggests UHRF1 transitions between a closed (apo) and open (ligand-bound) conformation, our binding data (new Figure 2 and Figure 3) shows that DNA – regardless of methylation status – stimulates histone binding activity and breaks intramolecular contacts between the SRA and TTD-PHD.

Unlike the conclusion drawn from qualitative binding measures by Feng et al., our quantitative data presented in Figure 2 and associated supplemental panels clearly shows that the histone- and DNA-binding activities are reciprocally regulated (e.g., DNA enhances histone binding and vice versa).

We show that the observed positive binding allostery *is not* HeDNA-specific. However, stimulation of ubiquitin ligase activity *is* HeDNA-specific (see Figure 4). We therefore conclude that an additional regulatory layer (beyond the model presented by Feng et al) accounts for HeDNA-induced ubiquitin ligase activity.

*The main concern I have with this manuscript regards their lysine prioritization model, for which I believe their current supporting evidence to be insufficient (see below).*

*Major concern:*

*1) In the Results section, the authors state: "UHRF1-dependent histone ubiquitylation was largely unaffected by free lysine in multi-turnover ubiquitylation assays, while UHRF1 auto-ubiquitylation was reduced (Figure 5)."*

*Looking at Figure 5 am not convinced that the presence of free lysine preferentially affects auto-ub over H3 peptide. If at all, auto-ub levels appear barely increased in the absence of free lysine, however, the di and mono-ub levels of H3 seem equally if not even more increased (cf. lanes 1 and 4 at di-ub or lanes 2 and 5 at mono-ub) suggesting that there is no prioritization of H3 lysines whatsoever. Nor is there any attempt to quantify this result.*

*They go on to say "We also confirmed this result using single turnover lysine discharge assays (Figure 5Figure 5—figure supplement 1)…" – yet these assays are in no way a confirmation of prioritization, as they simply show that H3 peptide ub can occur in the presence of free lysine, however, critically, they do directly compare rates H3 ub rates (normalized to auto-ub) in the absence vs. presence of free lysine as in Figure 5.*

*To me these results are much more consistent with the findings from Fang et al., and also shown here (Figure 3) whereby the combination of HeDNA, H3K9me3 and H3R2 unlock UHRF1 in an open confirmation associated with H3, facilitating efficient ubiquitylation of histone lysines. This more parsimonious possibility is even alluded to by the authors: Fifth paragraph of Discussion section – "We also cannot rule out the possibility that the increased rate of HeDNA-dependent ubiquitin transfer to lysines compared to auto-ubiquitylation or ubiquitin discharge to free lysine may arise from the UHRF1-bound histone lysines possessing ideal characteristics for accepting ubiquitin (e.g. spatially pre-arranged, protected from solvent, and/or appropriate pK)."*

*Overall, unless considerably improved evidence supporting the 'lysine prioritization' model is provided, this line of argument should be removed from the manuscript.*

We thank the reviewers for this thoughtful critique, and after consideration of this careful analysis and key results from several new experiments, we now propose that HeDNA functions with the TTD-PHD to direct the “substrate specificity” of UHRF1 ubiquitin ligase activity towards histone H3. Below we highlight experiments we’ve included in new versions of Figure 4 and Figure 5 that support our proposed model.

Figure 4: We carefully analyzed the rate of UHRF1 auto-ubiquitylation and peptide ubiquitylation in the presence of UnDNA and HeDNA. These results are striking: while HeDNA modestly enhances the rate of auto-ubiquitylation by 2.5-fold, the rate of peptide ubiquitylation in the presence of HeDNA is enhanced by more than 100-fold over the rate in the presence of UnDNA. Importantly, for UnDNA-bound UHRF1, the rate of auto-ubiquitylation is greater than 20-fold faster than the rate of peptide ubiquitylation, while in the presence of HeDNA, peptide ubiquitylation is favored (7-fold over). These results show that HeDNA-stimulated ubiquitin ligase activity is not generally activating RING activity. Coupled with competition and mutation analysis in Figure 4, results of ubiquitin discharge experiments in Figure 5 (see below), and mass spec analysis of ubiquitin target sites (Figure 6), we now feel sufficient evidence is provided to propose HeDNA functions with the TTD-PHD to direct the substrate specificity of the UHRF1 ubiquitylation reaction towards histone H3 (Figure 7).

Figure 5: In further support of our “substrate specificity” model, we show that E2~ub discharge is greatly enhanced only when it can be transferred to a histone substrate. We also show that both auto-ubiquitylation and free ubiquitin discharged to lysine is reduced when ubiquitin is transferred to a peptide.

*2) In my view the most important aspect of this paper is to establish the mechanism through which HeDNA binding stimulates ubiquitylation. This maybe partly explained by DNA binding disrupting the SRA-RING interaction with TTD-PHD (leading to enhanced binding to H3K9me), but this occurs with any DNA binding (modified or otherwise). So we are left, somewhat disappointingly, with the speculation that HeDNA binding causes specific conformational changes that reposition the RING domain to target the E2 ligase to particular H3 lysines.*

We absolutely agree with the reviewer that “establishing the mechanism” promoting HeDNA-dependent histone ubiquitylation is a high priority – particularly since we appear to have uncovered a distinct mechanism of RING activation. While we are now working to define the structural basis for these allosteric regulatory mechanisms controlling UHRF1 binding and enzymatic activities, these experiments are technically challenging and are not feasible to perform within the timescale of the revision. As the reviewers pointed out above, we hope the many layers of mechanistic advance presented in this manuscript in its current form will stimulate the community at large to further study (biochemically and structurally) mechanisms of allostery controlling the activities of chromatin regulators.